# Promoting Fairness Among Dynamic Agents in Online-Matching Markets under Known Stationary Arrival Distributions

**Will Ma**
Graduate School of Business
Columbia University
New York, NY 10027
`wm2428@gsb.columbia.edu`

**Pan Xu**
Department of Computer Science
New Jersey Institute of Technology
Newark, NJ 07102
`pxu@njit.edu`

## Abstract

Online (bipartite) matching under known stationary arrivals is a fundamental model that has been studied extensively with the objective of maximizing the total number of customers served. We instead study the objective of *maximizing the minimum matching rate across all online types*, which is referred to as long-run (individual) fairness. For Online Matching under long-run Fairness (OM-LF) with a single offline agent, we show that the first-come-first-serve (FCFS) policy is 1-competitive, *i.e.,* matching any optimal clairvoyant. For the general case of OM-LF: We present a sampling algorithm (SAMP) and show that (1) SAMP is of competitiveness of at least $1 - 1/e$ and (2) it is asymptotically optimal with competitiveness approaching one in different regimes when either all offline agents have a sufficiently large matching capacity, or all online types have a sufficiently large arrival rate, or highly imbalance between the total offline matching capacity and the number of online arrivals. To complement the competitive results, we show the following hardness results for OM-LF: (1) Any non-rejecting policy (matching every arriving online agent if possible) is no more than $1/2$-competitive; (2) Any (randomized) policy is no more than $(\sqrt{3} - 1)$-competitive; (3) SAMP can be no more than $(1 - 1/e)$-competitive suggesting the tightness of competitive analysis for SAMP. We stress that all hardness results mentioned here are independent of any benchmarks. We also consider a few extensions of OM-LF by proposing a few variants of fairness metrics, including long-run group-level fairness and short-run fairness, and we devise related algorithms with provable competitive performance.

## 1 Introduction

In online (bipartite) matching problems, nodes on one side of a bipartite graph are given in advance, while nodes on the other side arrive one-by-one. We refer to the two sets of nodes as *offline* and *online* agents, respectively. The edges incident to an online agent, which indicate the offline agents eligible to serve it, are revealed upon its arrival. An online matching algorithm must immediately serve each arriving agent using up to one eligible and unmatched offline agent; matches once made cannot be rearranged. The performance of an algorithm is typically evaluated as the total number of matches made, taking expectations as necessary if there is randomness in the arrivals or the algorithm. In this paper, we study online matching problems where performance is instead determined by the *fairness* in service provided to different online agent types.

**Online Matching under Long-Run Fairness** (OM-LF). Suppose there is a bipartite graph $G = (I, J, E)$, where $I$ and $J$ denote the sets of *types* for offline and online agents, respectively. Here each

38th Conference on Neural Information Processing Systems (NeurIPS 2024).

online type is defined based on certain attributes, *e.g.,* race and/or gender identity, which are observed upon arrival. For an offline agent (of type) $i \in I$, let $\mathcal{N}_i \subseteq J$ denote the "neighboring" online types that $i$ is eligible to serve. Similarly, for an online agent (of type) $j \in J$, let $\mathcal{N}_j \subseteq I$ denote the offline agent types eligible to serve $j$.[1] Each offline type $i \in I$ has an integer capacity $b_i \geq 1$ indicating the maximum number of online agents (with types in $\mathcal{N}_i$) that $i$ can serve.[2] (**Arriving Process**). Agents of each online type $j \in J$ arrive according to an *independent* Poisson process with *homogeneous* rate $\lambda_j > 0$, over a time horizon scaled to be $[0, 1]$.[3] When an online agent arrives in the time horizon [0,1], its type $j$ is revealed, and an online algorithm (or policy) ALG must make an immediate and irreversible decision from two options: either reject $j$ or serve $j$ by assigning it to an offline agent $i \in \mathcal{N}_j$ that still has available capacity. For the latter choice, $i$'s matching capacity will be reduced by one. (**Fairness Metric**). For each online type $j \in J$, let $X_j$ be the number of agents $j$ served by ALG and $A_j$ the number of $j$'s arrivals by the end of the time horizon. For any $\lambda > 0$, let $\mathrm{Pois}(\lambda)$ denote a Poisson random variable with mean $\lambda$. By our assumption, $A_j$ is distributionally identical to a $\mathrm{Pois}(\lambda_j)$ for every $j \in J$. Let $\mathcal{A} = (A_j)_{j \in J}$, which is referred to as the (random) *arrival vector* with each $A_j \sim \mathrm{Pois}(\lambda_j)$. We define a long-run fairness as follows.

$$\text{FAIR-L (Long-Run Fairness)} = \min_{j \in J} \frac{\mathbb{E}_{\mathcal{A}, \mathsf{ALG}}[X_j]}{\mathbb{E}_{\mathcal{A}}[A_j]} = \min_{j \in J} \frac{\mathbb{E}_{\mathcal{A}, \mathsf{ALG}}[X_j]}{\lambda_j}. \tag{1}$$

We aim to design an online algorithm ALG such that the achieved fairness is maximized. Note that all the information mentioned earlier, such as the graph $G = (I, J, E)$, the capacities of offline agents $\{b_i | i \in I\}$, and the arrival rates of online agents $\{\lambda_j | j \in J\}$, are accessible to the algorithm ALG as part of the input. *We refer to our problem as Online Matching under Long-Run Fairness (among online types)* (OM-LF).

**Remarks on** FAIR-L. (1) Random variables $\{X_j | j \in J\}$ are dependent on both the arrival vector $\mathcal{A}$ and any random bits used in the algorithm ALG itself. (2) For the long-run fairness FAIR-L, the time horizon typically represents one single day, and the algorithm is audited for fairness over a large number of days. In this case, the total number of agents (of type) $j$ served over all the days will be statistically close to $T$ times the numerator in (1), while the total number of arrivals of type $j$ over all the days will be statistically close to $T$ times the denominator. As a result, the audited performance is the minimum of this fraction over all types $j \in J$.

## 2    Preliminaries and Main Contributions

**Competitive Ratio**. Consider a given instance, characterized as
$\mathcal{I} = \big( G = (I, J, E), (b_i)_{i \in I}, (\lambda_j)_{j \in J} \big)$, an online algorithm ALG, and a given fairness metric. We overload notation and let $\mathsf{ALG}(\mathcal{I})$ denote the expected performance of ALG on $\mathcal{I}$. Similarly, we use $\mathsf{OPT}(\mathcal{I})$ to denote an optimal clairvoyant algorithm and its corresponding performance when the context is clear. Note that OPT can set the values of $(X_j)_j$ with advance knowledge of $\mathcal{A}$. With a fixed objective in mind, an algorithm ALG is said to be at least *c-competitive* if $\mathsf{ALG}(\mathcal{I}) \geq c \cdot \mathsf{OPT}(\mathcal{I})$ for any possible instance $\mathcal{I}$. The maximum possible value over $c \leq 1$ for which the above holds is called the *competitiveness* of ALG. The maximum possible competitiveness within a class of online algorithms is called the *competitive ratio* for that class.

**Holistic Nature of an Optimal Clairvoyant Algorithm**. Consider the classical (edge-weighted) online bipartite matching under KIID where the goal is to maximize the total weight of all matches. In that case, an optimal clairvoyant algorithm OPT will aim to optimize the objective on every realized instance and it can always find a deterministic strategy to do so. However, this may not be true in our problem. To see this, consider a simple example under FAIR-L where there is one single offline agent with $b = 1$ and two online types with $\lambda_1 = \epsilon$ and $\lambda_2 = 1$. For any realized arrival vector $\mathcal{A} = (A_1, A_2)$ with $A_1 \geq 1$ and $A_2 \geq 1$, we can show that the strategy of OPT on $\mathcal{A}$ can be characterized as follows: serve $j = 1$ and $j = 2$ with respective probabilities $p$ and $1 - p$, where

---

[1] In this paper, we refer to an agent of type $i$ ($j$) as agent $i$ ($j$) for simplicity when the context is clear.

[2] The capacity $b_i$ for offline type $i$ can be understood as the collective service capacity of all agents belonging to that particular type. In other words, it represents the total capacity available from serving agents of type $i$.

[3] Observe that our arrival setting is essentially equivalent to the *Know Identical Independent Distributions* (KIID), which is a discrete counterpart commonly assumed among studies of online matching under known distributions; see detailed discussions regarding "Online Bipartite Matching" in Section 3.

$p \geq (e-2)/(e-1) \approx 0.418$. This suggests that OPT has to resort to a randomized strategy on $\mathcal{A}$, and it does not suffice to simply maximize the objective of $\min\left(\mathbb{E}[X_1]/\epsilon, \mathbb{E}[X_2]/1\right)$ on every realization of $\mathcal{A}$. *More detailed discussions and clarifications can be seen in Appendix A.*

**Benchmark Linear Program (LP).** To guide the choice between offline agents, we write the following LP with variables $\{x_{ij} | (ij) \in E\}$ and $s$, where the variable $x_{ij}$ can be interpreted as the expected number of online type $j$ served by offline agent $i$ in an optimal clairvoyant algorithm (OPT), while $s$ can be interpreted as the "scale" of demand that can be served.

$$\max \ s \tag{2}$$

$$\sum_{j \in \mathcal{N}_i} x_{ij} \leq b_i \qquad \forall i \in I \tag{3}$$

$$\sum_{i \in \mathcal{N}_j} x_{ij} \geq s \cdot \lambda_j \ \ \forall j \in J \tag{4}$$

$$s, x_{ij} \geq 0 \qquad \forall(i,j) \in E \tag{5}$$

*Throughout this paper, we refer to the linear program described above as LP (2). We utilize the notation* OPT *to denote both an optimal clairvoyant policy and its corresponding performance when the context is clear.*

**Lemma 1.** LP (2) *is a valid benchmark for* OM-LF, *i.e., the optimal value of* LP (2), *denoted by $s^*$, is a valid upper bound for the performance of a clairvoyant algorithm. Therefore,* OPT $\leq \min\{s^*, 1\}$.

Note that it is important in Lemma 1 that we also upper bound OPT by 1; this will allow us to later establish asymptotic optimality in $s^*$.

*Proof.* Consider any clairvoyant algorithm OPT. Let $X_{ij}$ be the random variable for the number of times it uses $i$ to serve $j$, with $X_j = \sum_{i \in \mathcal{N}_j} X_{ij}$. Recall that OPT $= \min_{j \in J} \mathbb{E}[X_j]/\lambda_j$. It can be checked that setting $x_{ij} = \mathbb{E}[X_{ij}], s = $ OPT constitutes a feasible LP solution with objective value OPT. Therefore, LP $\geq$ OPT, and $1 \geq$ OPT holds by definition, completing the proof. $\square$

**Remarks on** LP (2). Among existing studies of online matching under known distributions, all benchmark LPs are designed solely for outputting an upper bound on the performance of an optimal clairvoyant (OPT), which is then used to establish a lower bound on the resulting competitiveness by comparing the performance of an online algorithm against it. In contrast, the benchmark LP (2) proposed here serves dual purposes. The optimal value $s^*$ of LP (2), not only offers an upper bound on OPT but also plays a crucial role in scaling online sampling distributions (*e.g.,* in Algorithm SAMP of Theorem 1). This means that benchmark LP (2) actively participates in the online algorithm design, serving as a key component in shaping the online decision-making process of the algorithms.

## 2.1 Main Contributions

In this paper, we introduce a fairness metric among online types, defined in (1), and propose a model, called *online matching under long-run fairness among online types* (OM-LF). Our contributions are summarized as follows.

### 2.1.1 A Warm-Up for OM-LF with a Single Offline Agent.

We observe that when there is a single offline agent, the optimal online algorithm is First-Come-First-Serve (FCFS), which assigns all incoming agents to the offline agent as long as capacity is available. We demonstrate that FCFS is 1-competitive.

**Proposition 1.** *For* OM-LF *with one single offline agent,* FCFS *is 1-competitive, making it optimal among all algorithms.*

*Proof.* Suppose that $I$ consists of a single offline agent with capacity $b$. Let $A$ be the random variable for the total number of online arrivals, in which case FCFS serves the first $\min\{A, b\}$ arrivals. Conditioned on any value $A > 0$, the distribution of online types served is proportional to the

arrival rates $\lambda_j$. That is, for any online type $j \in J$, the expected number of type-$j$ agents served is $\mathbb{E}\Big[ \min\{\mathrm{Pois}(\sum_{j \in J} \lambda_j), b\}\Big] \cdot \frac{\lambda_j}{\sum_{j \in J} \lambda_j}$. All in all, FCFS achieves a fairness of $\mathbb{E}\Big[ \min\{\mathrm{Pois}(\sum_{j \in J} \lambda_j), b\}\Big] / \sum_{j \in J} \lambda_j$, which cannot be beaten even by a clairvoyant algorithm since the total number of agents served cannot exceed $\mathbb{E}\Big[ \min\{\mathrm{Pois}(\sum_{j \in J} \lambda_j), b\}\Big]$. This shows that FCFS is 1-competitive and is also the optimal clairvoyant algorithm. $\qquad\square$

### 2.1.2 General Cases of OM-LF.

We consider OM-LF with multiple offline agents, where each offline agent may have a different capacity.

**Theorem 1** (Section 4)**.** *There is an algorithm (*SAMP*) for OM-LF, whose competitiveness is lower-bounded by $\kappa(b, s^*) \doteq \mathbb{E}[\min(\mathrm{Pois}(b/s^*), b)] \cdot \max(s^*, 1)/b \geq \kappa(1, 1) = 1 - 1/\mathsf{e}$, where $b = \min_{i \in I} b_i$ and $s^* \in (0, \infty)$ is the optimal value to benchmark LP (2) that measures the inverse of the overall demand saturation in the system. Meanwhile, the competitiveness of* SAMP *approaches* 1 *when either $b \to \infty$ or $s^* \to 0^+$ (demand dominates supply) or $s^* \to \infty$ (supply dominates demand).*

**Theorem 2** (Section 5)**.** *For OM-LF, the following hardness results hold: (1) Any non-rejection algorithm (possibly randomized) that serves an incoming agent whenever possible is no more than $1/2$-competitive. (2) Any algorithm (possibly randomized) is no more than $(\sqrt{3} - 1)$-competitive. (3) The competitive analysis of* SAMP *is tight, as it cannot be more than $(1 - 1/\mathsf{e})$-competitive. All the hardness results mentioned in (1), (2), and (3) are independent of any benchmarks.*[4]

### 2.1.3 Extension of OM-LF to Group-Level Fairness.

We consider an extension of OM-LF when each online type belongs to some pre-defined protected groups. Specifically, suppose there is a collection of protected groups $\mathcal{G}$, where each group $g \in \mathcal{G}$ is a subset of $J$, indicating the online agent types that fall under group $g$. We assume w.l.o.g. that every type $j \in J$ is contained in at least one group (otherwise we could discard and never serve that type); note however that groups can overlap with each other. We generalize long-run fairness (FAIR-L), as defined in Equation (1), to a group-level version with respect to groups of $\mathcal{G}$ as follows:

$$\text{FAIR-L}(\mathcal{G}) = \min_{g \in \mathcal{G}} \frac{\mathbb{E}_{\mathcal{A}, \mathsf{ALG}}[X(g)]}{\sum_{j \in g} \lambda_j}, \tag{6}$$

where $X(g) = \sum_{j \in g} X_j$ denotes the (random) number of types in $g$ served in an algorithm ALG.

**Comparison between (Individual) Long-Run Fairness** FAIR-L **in** (1) **and Group-Level Long-Run Fairness in** (6)**.** The original long-run fairness FAIR-L, as defined in (1), can be considered a special case of group-level long-run fairness in (6) with respect to $\mathcal{G} = \{g = \{j\} | j \in J\}$, where each group consists of a single online type. Therefore, FAIR-L in (1) can be interpreted as *individual* long-run fairness with respect to every single type, as opposed to group-level fairness with respect to a pre-defined set of groups $\mathcal{G}$. We emphasize that overlaps among groups can potentially doom classical policies, such as first-come-first-serve (FCFS), even under very simple settings. Proposition 1 states that FCFS is 1-competitive (i.e., matching the performance of a clairvoyant optimal) for individual long-run fairness when there is a single offline agent. In contrast, Example 1 (see below) demonstrates that *FCFS is zero-competitive for group-level long-run fairness, as defined in (6), under the same setting of a single offline agent.*

**Example 1** (FCFS is zero-competitive for group-level fairness)**.** *Consider the following example: There is a single server with unit capacity. There are $n + 1$ online types, indexed as $j = 0, 1, 2, \ldots, n$, each with an arrival rate of 1, and $n$ groups such that each group $k = 1, 2, \ldots, n$ consists of two types $(0, j)$ with $j = k$. We can verify the following: (1) Any clairvoyant optimal (*OPT*) can achieve a group-level (long-run) fairness of at least $(1 - 1/e)/2$. For any offline policy prioritizing serving arriving online types of $j = 0$, it achieves a group-level fairness of at least $(1 - 1/e)/2$. (2) FCFS achieves a group-level fairness of $1/(n + 1)$: Note that each group has one agent served by FCFS*

---

[4]When we state that all hardness results provided in the paper are independent of any benchmarks, we mean that all competitiveness results are computed directly against the performance of a clairvoyant optimal policy (OPT), rather than any upper bound on OPT (e.g., the optimal value of a benchmark LP, as claimed in Lemma 1).

*only when the first arriving agent belongs to one of the two types in that group, which occurs with probability $2/(n+1)$. Thus, we conclude that FCFS is zero-competitive for group-level fairness (when $n \to \infty$).*

**Theorem 3** (Appendix G)**.** *For* OM-LF *with group-level long-run fairness: (1) There exists an algorithm (*SAMP-G*) that achieves a competitive ratio of at least $1 - \mathrm{e}^{-b}b^b/b!$ with $b = \min_{i \in I} b_i$, which is increasing over $b \in \{1, 2, \ldots\}$ and approaches 1 as $b \to \infty$; (2) There exists an algorithm (*RESERVE*) that achieves a competitive ratio of at least $1 - \mathrm{e}^{-\lambda}\lambda^\lambda/\lambda!$ with $\lambda = \min_{j \in J} \lambda_j$, which approaches 1 as $\lambda \to \infty$.*

**Remarks on Results in Theorems 1, 2, and 3**. (1) Ma et al. [28] considered both long-run individual and group-level fairness maximization, but their focus was on fairness among *offline agents*. This is in contrast to the emphasis on fairness among online types. Another difference is that the work of [28] assumed integral arrival rates among online types and utilized this assumption to propose a strengthened benchmark LP. Additionally, they claimed that each online type could be made to admit a unit arrival rate ($\lambda_j = 1$) by creating multiple copies. In our paper, however, we do not make any assumptions regarding the arrival rates among online types: They can take any fractional or integer values, allowing for a more general analysis of fairness among online types. A more detailed discussion can be seen in Appendix B. (2) As noted before, our arrival setting is essentially equivalent to the *Know Identical Independent Distributions* (KIID), which is a discrete arrival setting commonly assumed in the study of online matching under known distributions. For Online Matching under KIID (OM-KIID), the most commonly studied objective is the maximization of the total weight of all matches under different weight settings, including unweighted, vertex-weighted (offline-side), and edge-weighted scenarios. To date, there have been only two known hardness results for OM-KIID with general arrival rates: one is $0.823$ for unweighted and vertex-weighted due to [31] and the other is $0.703$ for edge-weighted due to [21]. Our hardness result of $\sqrt{3} - 1 \approx 0.732$ contributes to this short list of hardness results for OM-KIID. Notably, our analysis focuses on the objective of maximizing long-run fairness among online types, which adds a new dimension to the study of OM-KIID and expands the understanding of the inherent challenges and limitations in achieving fairness in online matching scenarios.

#### 2.1.4 Another Fairness Metric: Short-Run Fairness.

We propose a second fairness metric, called *Short-Run Fairness*, which is defined as follows:

$$\text{FAIR-S} = \mathbb{E}_{\mathcal{A}} \left[ \min_{j \in J: A_j > 0} \frac{\mathbb{E}_{\mathsf{ALG}}[X_j | \mathcal{A}]}{A_j} \right], \tag{7}$$

where $\mathcal{A} = (A_j)_{j \in J}$ is the (random) arrival vector with $A_j \sim \text{Pois}(\lambda_j)$ being the number of arrivals of type $j \in J$.[5]

**Remarks on** FAIR-S. (a) In the numerator of FAIR-S, $\mathbb{E}_{\mathsf{ALG}}[X_j | \mathcal{A}]$ is a conditional expectation taken over only the randomness in the algorithm ALG. (b) In FAIR-S, types $j$ with no realized arrivals (for which the denominator $A_j = 0$) are ignored. Also, we assume that FAIR-S $= 1$ in case all $A_j = 0$, *i.e.,* no online agents arrive. (c) No inherent relation can be imposed on FAIR-L and FAIR-S. *There are examples supporting both possibilities that* FAIR-L $>$ FAIR-S *and* FAIR-L $<$ FAIR-S; *see details in Appendix C*. (d) For the short-run fairness, the algorithm is audited for fairness based on the realized arrivals every single day. To avoid impossibility results, evaluation in the numerator of (7) is based on the *expected* service over any randomness in the algorithm.[6] Interpreted another way, when evaluating Short-Run Fairness, we are allowing for *fractional* allocations to be made on a given day. The overall performance (7) then takes the expectation of the daily audit scores over a large number of days. (e) Note that the definitions of long-run and short-run fairness, as

---

[5]Note that we can define the group-level short-run fairness following the same way as FAIR-L: FAIR-S($\mathcal{G}$) $= \mathbb{E}_{\mathcal{A}} \left[ \min_{g \in \mathcal{G}: A(g) > 0} \frac{\mathbb{E}_{\mathsf{ALG}}[X(g) | \mathcal{A}]}{A(g)} \right]$, where $A(g) = \sum_{j \in g} A_j$ denotes the number of arrivals of types in $g$, and $X(g) = \sum_{j \in g} X_j$ the number of types in $g$ served by ALG. We can verify that all the analysis and results obtained for FAIR-S in this paper, as shown in Theorem 4, also apply to group-level short-run fairness for any collection of groups $\mathcal{G}$.

[6]Observe that any deterministic algorithm will yield fairness of zero during peak hours when the total number of online agent arrivals significantly exceeds the serving capacity of offline agents.

shown in equations (1) and (7) respectively, bear similarities to two other concepts known as *ex-ante* and *ex-post* fairness. These concepts have been extensively studied in the field of online resource allocation [2, 15, 30]. Specifically, our notion of long-run fairness aligns more closely with the idea of ex-ante fairness, which focuses on the minimum expectation, while the short-run fairness aligns more closely with the concept of ex-post fairness, which emphasizes the expectation of the minimum outcome.

Unlike FAIR-L, online matching under FAIR-S is quite technically challenging, even for upper-bounding the performance of an optimal clairvoyant policy (OPT). So far, we have not found any appropriate linear program that can serve as a valid benchmark for OPT as we did for FAIR-L. That being said, we take an initial stab by focusing on a simple case when there is a *single* offline agent with a service capacity of $b$. Even in this special case, characterizing the optimal online algorithm that maximizes FAIR-S is technically challenging. This contrasts with Proposition 1, which states that FCFS is 1-competitive under FAIR-L with a single offline agent.

**Theorem 4** (Appendix H). *For online matching under* FAIR-S *with a single offline agent of capacity* $b$ *and a total online arrival rate of* $\lambda := \sum_{j \in J} \lambda_j$: *(1)* FCFS *is* $0.863$-*competitive when* $\lambda \leq 1$; *(2) No algorithm can achieve a competitive ratio greater than* $0.942$ *when* $b = \lambda = 1$; *(3) There exists an algorithm (*Prob-Rej*) that achieves a competitive ratio of at least* $1 - o(1)$, *where* $o(1)$ *is a vanishing term as* $\lambda \to \infty$.

## 3 Other Related Work

**Online Bipartite Matching**. Online bipartite matching was pioneered by Karp et al. [24] and its variants have gained interest during the past two decades in the CS community. Based on the arrival setting of online agents, there are three major categories: (1) Adversarial, the arrival sequence is fully unknown but fixed, see, *e.g.,* [7, 33]; (2) Random arrival order, the full arrival sequence forms a random permutation over a set of unknown agents, see, *e.g.,* [36, 29, 23, 17, 13]; (3) known/unknown distributions, the stochastic arrivals of online agents follow certain known/unknown distributions. A special case here is when online arrivals follow Known Independent and Identical Distributions (KIID), see, *e.g.,* [14, 18, 31, 22]. Our arrival setting shares the spirit of KIID, though we consider a continuous version instead of discrete. Huang and Shu [19] considered the same arrival setting as ours and show that under mild assumptions,[7] the performance of an online algorithm is almost the same under the two arrival settings (*i.e.,* KIID and independent Poisson process).

There is an interesting connection between our model under Long-Run fairness and the *online-side* vertex-weighted online matching under KIID. So far, studies about vertex-weighted online matching all focus on the setting of the offline side, *i.e.,* all edges incident to any given offline agent share a weight. Examples include [19] and [6] under KIID, [20] under random arrival order, and [1] under adversarial arrival order. By contrast, we believe that our analysis and results can be applied to the *online-side* vertex-weighted online matching problem, which we leave as future work.

**Fair Operations**. Fairness in operations is a topic of increasing interest and we aim to provide a brief literature review. Classical works in this area include [4] and [5] which define the price of fairness and efficiency-fairness tradeoff, respectively, in an axiomatic fashion. Gig platforms have motivated many studies on balancing multiple objectives [26], including fair allocation on the rider side [34] and income equality on the driver side [38] in rideshare. Fair *pricing* to the customer side has been more generally studied in [12], while fair allocation has been studied in transportation problems [10, 9, 35] and COVID-19 vaccine distributions [37]. We note that in the application of [10], the authors justify prioritizing transportation for certain groups (e.g. seniors), instead of balancing fairness across all groups as we do.

More generally, online resource allocation frameworks that can capture fairness have been considered in [3, 25, 11]. These papers all derive regret bounds which are sublinear in the number of arrivals, while we derive competitive ratio bounds which hold universally and establish asymptotic optimality

---

[7]Specifically, as stated in the paper [19], for any algorithm, its competitiveness can be translated between the two models up to a multiplicative factor of $1 - O(\Lambda^{-1/2})$, where $\Lambda := \sum_{j \in J} \lambda_j$ represents the total arrival rate of all online types. Note that, it is a common practice to consider $\Lambda \to \infty$ or $n \to \infty$ (the counterpart of $\Lambda$ in the KIID setting) in competitive analysis for online matching models under known distributions, which is assumed in this paper as well.

in regimes (involving the demand saturation) not previously captured. However, we should note that our techniques appear to be reliant on the max-min objective function, while these papers allow for more general functions.

# 4 Proof of Theorem 1: Online Matching Under Long-Run Fairness (OM-LF)

## 4.1 Algorithm SAMP and Intuitions

In this section, we present an LP-based sampling algorithm, denoted by SAMP, which is $(1 - 1/e)$-competitive and asymptotically optimal in many parameter regimes. Let $\{x_{ij}^*, s^*\}$ be an optimal solution to the benchmark LP (2). For all $j \in J$, WLOG assume that $x_j^* \doteq \sum_{i \in \mathcal{N}_j} x_{ij}^* = s^* \cdot \lambda_j$.[8] SAMP is formally stated in Algorithm 1.

---
**ALGORITHM 1:** An LP-based Sampling Algorithm (SAMP)
---
1   Solve LP (2) to get an optimal solution $\{x_{ij}^*, s^*\}$.
2   Let an online agent (of type) $j$ arrive at time $t$.
3   Sample a neighbor $i \in \mathcal{N}_j$ with probability $x_{ij}^*/(s^* \cdot \lambda_j)$.
    /* This is a valid distribution since $\sum_{i \in \mathcal{N}_j} x_{ij}^*/(s^* \cdot \lambda_j) = x_j^*/(s^* \cdot \lambda_j) = 1$.       */
4   If $i$ is safe, *i.e.*, $i$ has remaining capacity, then assign $i$ to serve $j$; otherwise, reject $j$.

---

SAMP does not re-sample an offline agent if the first one sampled is unavailable, so it does not share the property of FCFS that an incoming agent is served whenever possible. The property that SAMP sometimes "rejects" an incoming agent is imperative for it to surpass the barrier of $1/2$, as suggested by Theorem 2.

## 4.2 Proof of Theorem 1: Competitive Analysis of SAMP

First, we use two lemmas to analyze the number of times each online type is served by SAMP.

**Lemma 2.** *For each* $i \in I$ *and* $t \in [0, 1]$, *let* $\mathsf{SF}_{it}$ *indicate if offline agent* $i$ *is safe at the instantaneous point in time* $t$ *in algorithm* SAMP, *i.e.,* $i$ *still has remaining capacity at* $t$. $\mathbb{E}[\mathsf{SF}_{it}] \geq \Pr[\mathrm{Pois}(b_i t/s^*) < b_i]$, *for all* $i \in I$ *and* $t \in [0, 1]$.

*Proof.* An offline agent $i$ is safe at time $t$ if and only if there have been fewer than $b_i$ arrivals before $t$ which sampled $i$. Such arrivals are Poisson with total rate $\sum_{j \in \mathcal{N}_i} \lambda_j \cdot \frac{x_{ij}^*}{s^* \cdot \lambda_j}$, which is at most $b_i/s^*$ by LP constraints (3). Therefore, the number of such arrivals is Poisson with mean at most $b_i \cdot t/s^*$, completing the proof.     □

**Lemma 3.** *Let* $X_j^S$ *be the random number of times type* $j$ *is serviced in* SAMP. *Then for all* $j \in J$,

$$\frac{\mathbb{E}[X_j^S]}{\lambda_j} \geq s^* \cdot \min_{i \in I} \frac{\mathbb{E}[\min\{\mathrm{Pois}(b_i/s^*), b_i\}]}{b_i}. \tag{8}$$

*Proof.* Consider any $i, j$ for which an offline agent $i$ is eligible to serve online type $j$. Let $X_{ij}^S$ be the random variable for the number of times SAMP uses $i$ to serve $j$. $X_{ij}^S$ is incremented whenever: (1) type $j$ arrives (occurring following Poisson process of rate $\lambda_j$); (2) $i$ is sampled (occurring with probability $x_{ij}^*/(s^* \cdot \lambda_j)$); and (3) $i$ is safe (occurring with probability at least $\Pr[\mathrm{Pois}(b_i t/s^*) < b_i]$, by Lemma 2). Since these events are mutually independent, we have

$$\mathbb{E}[X_{ij}^S] \geq \int_0^1 \lambda_j \cdot \frac{x_{ij}^*}{s^* \cdot \lambda_j} \cdot \Pr[\mathrm{Pois}(b_i \cdot t/s^*) < b_i] dt$$

$$= \frac{x_{ij}^*}{b_i} \int_0^1 \frac{b_i}{s^*} \cdot \Pr[\mathrm{Pois}(b_i \cdot t/s^*) < b_i] dt = \frac{x_{ij}^*}{b_i} \cdot \mathbb{E}[\min\{\mathrm{Pois}(b_i/s^*), b_i\}].$$

---

[8]This is because if $\sum_{i \in \mathcal{N}_j} x_{ij}^* > s^* \cdot \lambda_j$, then we can re-scale the values of $x_{ij}^*$ by $(s^* \cdot \lambda_j)/(\sum_{i \in \mathcal{N}_j} x_{ij}^*)$, without violating feasibility.

The final equality holds because the integral "counts" an arrival from a Poisson process of rate $b_i/s^*$ whenever the number of arrivals thus far is less than $b_i$; this equals, in expectation, the number of arrivals from such a process truncated by $b_i$.

Now, for any online type $j \in J$, let $X_j^S = \sum_{i \in \mathcal{N}_j} X_{ij}^S$ be the random variable for the number of times SAMP serves $j$. The previous derivation for $X_{ij}^S$ implies that

$$\mathbb{E}[X_j^S] \geq \sum_{i \in \mathcal{N}_j} x_{ij}^* \cdot \frac{\mathbb{E}[\min\{\text{Pois}(b_i/s^*), b_i\}]}{b_i} \geq s^* \cdot \lambda_j \cdot \min_{i \in \mathcal{N}_j} \frac{\mathbb{E}[\min\{\text{Pois}(b_i/s^*), b_i\}]}{b_i},$$

where the second inequality uses LP constraint (4). This completes the proof. $\qquad\square$

Having derived the expression on the RHS of (8), we aim to lower bound it in terms of simpler expressions of $b_i$ and $s^*$. Recall that $\kappa(b, s) \doteq \max\{s, 1\} \cdot \frac{\mathbb{E}[\min\{\text{Pois}(b/s), b\}]}{b}$ for any integer $b \geq 1$ and $s > 0$. For any $\lambda > 0$ and $s > 0$, define $\eta(\lambda, s) = \frac{\mathbb{E}[\min \text{Pois}(\lambda), \lambda s]}{\lambda \cdot \min(s, 1)}$, a related function we will later use in our analysis. We can verify that $\kappa(b, s) = \eta(b/s, s)$ and $\eta(\lambda, s) = \kappa(\lambda s, s)$. Below are a few properties of $\kappa(b, s)$.

**Lemma 4** (Appendix D). *(1) For any fixed $s > 0$, $\kappa(b, s)$ is increasing over $b \in \{1, 2, \ldots\}$; (2) For any fixed integer $b \geq 1$, $\kappa(b, s)$ is minimized at $s = 1$; (3) For all integers $b \geq 1$ and $s > 0$, $\kappa(b, s) \geq \kappa(1, 1) = 1 - 1/e$; (4) When $s > 1$, $\kappa(b, s) \geq 1 - \exp\big(-b \ln s \cdot (1 - o(1))\big)$, where $o(1)$ vanishes when $s \to \infty$; (5) When $s = 1$, $\kappa(b, 1) \geq 1 - \frac{1}{\sqrt{2\pi(b-1)}}$ with $b > 1$; (6) When $0 < s < 1$, $\kappa(b, s) \geq 1 - \exp\big(-\frac{b}{2s}(1 - s)^2\big)$.*

***Proof of Theorem 1.*** By Lemma 3, the fairness of SAMP under FAIR-L is at least

$$\frac{\mathbb{E}[X_j^S]}{\lambda_j} \geq s^* \cdot \min_{i \in I} \frac{\mathbb{E}[\min\{\text{Pois}(b_i/s^*), b_i\}]}{b_i} \geq s^* \cdot \frac{\mathbb{E}[\min\{\text{Pois}(b/s^*), b\}]}{b},$$

where $b = \min_i b_i$, and the last inequality follows from Part (1) of Lemma 4. By Lemma 1, OPT $\leq \min\{s^*, 1\}$. Putting these statements together, we see that the competitive ratio is lower bounded by

$$\frac{s^*}{\min\{s^*, 1\}} \cdot \frac{\mathbb{E}[\min\{\text{Pois}(b/s^*), b\}]}{b} = \kappa(b, s^*).$$

All of the properties about $\kappa(b, s^*)$ follow directly from Lemma 4, with the asymptotic behavior when $b \to \infty$, $s^* \to 0^+$, or $s^* \to \infty$ following from the bounds given in parts (4)–(6) of Lemma 4. $\qquad\square$

## 5 Proof of Theorem 2

**Example 2** (**Bad Example**). *$J$ consists of a large number of "rare types" $t = 1, \ldots, n$ each with $\lambda_t = 1/n$ and a single "common type" 0 with $\lambda_0 = n - 1$. $I$ consists of $n$ unit-capacity servers such that each rare type $t = 1, \ldots, n$ can only be served by a server $t$, but all servers can serve the common type. The graph structure is shown in Figure 1.*

*We can verify that the optimal clairvoyant algorithm gives priority to rare types, and uses each server $t \in \{1, 2, \ldots, n\}$ for which type $t$ never arrived to serve the common type. The expected amount of each rare type $t$ served is $1 - e^{-1/n} \geq 1/n - O(1/n^2)$, while the expected number of the common type served is at least $n - 1 - n(1 - e^{-1/n}) \geq n - 2$. Thus, we claim any optimal clairvoyant (OPT) can achieve a long-run fairness of $1 - O(1/n)$ under FAIR-L.*

### 5.1 Proof of Part (1) of Theorem 2: $1/2$-Upper Bound for Non-Rejecting

We first use Example 2 to show that non-rejecting algorithms cannot be better than $1/2$-competitive.

**Lemma 5.** *On Example 2, any non-rejecting online algorithm is no more than $1/2$-competitive relative to the best clairvoyant algorithm.*

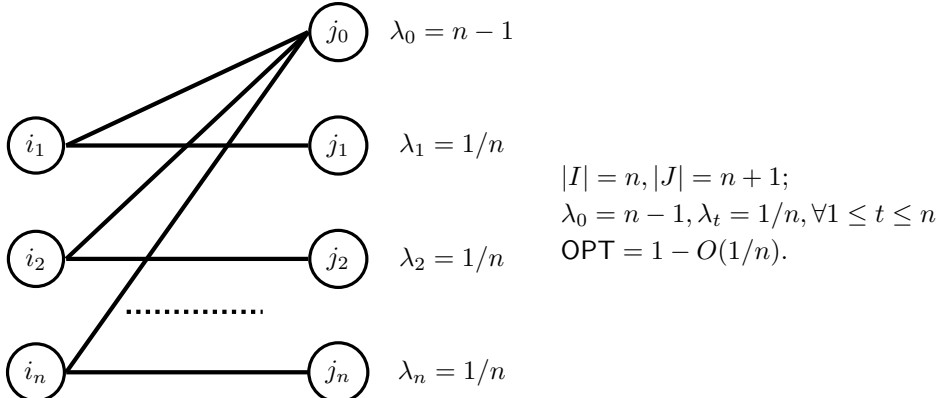

Figure 1: A bad example used to show hardness results for any randomized and non-rejecting algorithms and the tightness of competitive analysis for SAMP.

*Proof.* An online algorithm that serves incoming agents whenever possible must have a (randomized) order for available servers to use on the common type. The rare type which is in position $P \in [n]$ in this order must have an arrival before the $P$'th arrival of the common type 0, to have any chance of being served. For a given rare type $t$, let $P_t$ denote the (randomized) position of type $t$ in this order. For any position $P \in [n]$, let $\mathrm{Arr}(P) \in [0, 1]$ denote the arrival time of the $P$'th arrival of the common type 0. By independence of the Poisson processes for the arrivals of different types, the probability of a rare type $t$ being served is at most

$$\mathbb{E}[1 - \exp(-\mathrm{Arr}(P_t)/n)] \leq \mathbb{E}[\mathrm{Arr}(P_t)/n], \tag{9}$$

which in turn is at most $\left(\frac{\mathbb{E}[P_t]+1}{n}\right)/n$ for sufficiently large $n$.[9] Recall that for each rare type $t = 1, 2, \ldots, n$, $P_t \in \{1, 2, \ldots, n\}$ denotes the position of type $t$ in a randomized order adopted by a non-rejecting policy. Thus, we claim that $\sum_{t=1}^{n} \mathbb{E}[P_t] = n(n+1)/2$. This implies that at least one rare type $t$ must satisfy $\mathbb{E}[P_t] \leq (n+1)/2$. Therefore, the fairness of this online algorithm cannot exceed $1/2 + O(1/n)$. $\qquad\square$

**Remarks**. In Appendix E, we present a proof of Lemma 6, which can be viewed as a stronger version of Lemma 5. The alternative proof, though more technically involved, provides a foundational framework for analyzing a much broader class of online policies and is more self-explanatory. Additionally, in Appendix F, we focus on the bad example (Example 2) to conduct a quantitative study on the tradeoff between competitiveness and the number of rejected agents.

## 5.2 Proof of Part (2) of Theorem 2: $(\sqrt{3} - 1)$-Upper Bound for Any Randomized

*Proof.* On Example 2, any online algorithm that is going to reject the common type is better off doing so sooner rather than later, since an earlier rejection allows more time to observe which rare types arrive, and give those types priority. For any $\tau \in [0, 1]$, suppose that the online algorithm, denoted by $\mathsf{ALG}(\tau)$, starts accepting common types after time $\tau$.

The online algorithm must have some (possibly randomized) order of offline servers to use when it wants to serve the common type. The rare type whose corresponding offline server is in position $P \in [n]$ in this order must have an arrival before the $P$'th arrival of the common type *after time* $\tau$, to have any hope of being served. Counting from time $\tau$, the $P$'th arrival of the common type will occur before $\tau + \frac{P+1}{n}$ w.h.p. as $n \to \infty$. As a result, the probability of this rare type being served is at most

$$1 - \exp\left(-\frac{\min\{\tau + \frac{P+1}{n}, 1\}}{n}\right) \leq \frac{\min\{\tau + \frac{P+1}{n}, 1\}}{n}.$$

As $n \to \infty$, the average value of the RHS expression over $P = 1, \ldots, n$ is

$$\frac{1}{n} \int_0^1 \min\{\tau + z, 1\} dz = \frac{1}{n}\left(\tau + \frac{1}{2} - \frac{1}{2}\tau^2\right).$$

---

[9]This is because as $n \to \infty$, the arrivals of a Poisson process of rate $n$ are evenly spaced in $[0, 1]$ w.h.p.

Therefore, even using a randomized order, there must exist a rare type whose probability of being served is at most $\frac{1}{n}(\tau + \frac{1}{2} - \frac{1}{2}\tau^2)$. Meanwhile, for any $\tau$, the expected number of common types served can be at most $(n-1)(1-\tau)$. Since the arrival rates for rare and common types are $\frac{1}{n}$ and $n-1$ respectively, the fairness of the online algorithm cannot exceed $\min\{\tau + \frac{1}{2} - \frac{1}{2}\tau^2, 1-\tau\}$.

We can verify that the fairness of the online algorithm is maximized at $\tau = 2 - \sqrt{3}$, in which case it equals $\sqrt{3} - 1$. Meanwhile, for Example 2, a clairvoyant algorithm can achieve a fairness of 1. This completes the proof. $\qquad\square$

## 5.3  Proof of Part (3) of Theorem 2: Tightness of Competitive Analysis of SAMP

*Proof.* For each $t \in \{1, 2, \ldots, n\}$ and $\tilde{t} \in \{0, 1, 2, \ldots, n\}$, let $x^*_{t,\tilde{t}}$ denote the value set by an optimal solution of LP (2) on the edge $(i_t, j_{\tilde{t}})$ in Example 2. We can verify that the optimal LP solution sets $x^*_{t,t} = 1/n$ and $x^*_{t,0} = 1 - 1/n$ for each $t = 1, \ldots, n$, and $s^* = 1$. As a result, each offline agent $t \in [n]$ serves a demand with a total arrival rate of one. Thus, each offline agent $t$ successfully serves a demand with probability $1 - 1/e$ since no demand arrives otherwise (which occurs with probability $1/e$). Conditioned on offline agent $t$ serving a demand, the probability of that demand being of rare type $t$ (instead of the common type 0) is $1/n$. Therefore, for any rare type $t \in [n]$, we have $\mathbb{E}[X_t]/\lambda_t = 1 - 1/e$, where $X_t$ denotes the random number of times type $t$ is serviced. Thus, and under FAIR-L, SAMP achieves a fairness of at most $1 - 1/e$. Meanwhile, in Example 2, we show that a clairvoyant algorithm can achieve a FAIR-L of $1 - O(1/n)$, completing the proof. $\quad\square$

## 6  Conclusion and Reservations

We propose algorithms for maintaining statistical parity in the service rates provided to different online types or groups when agents arrive dynamically. We believe this has the potential to make a positive impact on *e.g.,* sharing economy platforms, where our algorithms will give priority to under-served groups when matching agents, thereby boosting their rates of service. However, we admit that our algorithms do not address any underlying discrimination issues of why those groups were less commonly served by hosts/drivers in the first place. Also, our algorithms are only "fair" with respect to the fairness metrics we defined: Our model does not necessarily guarantee fairness over all arriving individuals. Relatedly, our algorithms could have the negative consequence of causing "unfairness" by violating the first-come-first-serve principle, since sometimes earlier-arriving agents are rejected in order to preserve capacity for later-arriving agents who may belong to protected groups.

## Acknowledgments and Disclosure of Funding

Pan Xu was partially supported by NSF CRII Award IIS-1948157. The authors gratefully acknowledge the valuable comments provided by the anonymous reviewers.

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

# A Further Discussions on the Holistic Nature of an Optimal Clairvoyant

We emphasize that in our problem, any clairvoyant optimal policy (OPT) needs to consider the entire input instance as a whole when optimizing its decisions for each specific arrival sequence of online agents. (Recall that any clairvoyant optimal policy has the advantage of optimizing its decisions after observing the full arrival sequence.) This stands in contrast to clairvoyant optimal policies for existing, well-studied online matching problems when the objectives are to maximize the (expected) total weight of all matches. In these cases, any clairvoyant optimal policy only needs to optimize its decisions for each specific arrival sequence of online agents *without* considering any information or consequences of other possible arriving sequences.

Now, let us revisit the toy example of OM-LF, as mentioned before, where there is one single offline agent with $b = 1$ and two online types with $\lambda_1 = \epsilon$ and $\lambda_2 = 1$. According to the arrival setting, agents of $j = 1$ and $2$ each arrive following an independent Poisson process of rate 1 and $\epsilon$, respectively. OPT expects the following four possible scenarios. Let $A_1$ and $A_2$ denote the numbers of arrivals of $j = 1, 2$, respectively.

- Case 1: $A_1 = 0, A_2 \geq 1$, which occurs with probability equal to $e^{-\epsilon}(1 - 1/e)$;
- Case 2: $A_1 \geq 1, A_2 = 0$, which occurs with probability equal to $(1 - e^{-\epsilon})(1/e)$;
- Case 3: $A_1 \geq 1, A_2 \geq 1$, which occurs with probability equal to $(1 - e^{-\epsilon})(1 - 1/e)$;
- Case 4: $A_1 = 0, A_2 = 0$, which occurs with probability equal to $e^{-\epsilon - 1}$.

Observe that since we have one single offline agent $i$ with a unit capacity, OPT has one single choice in Cases 1, 2, and 4. For example, in Case 1, OPT will match one arrival of $j = 2$ with $i$, and in Case 2, OPT will match any arrival of $j = 1$ with $i$, and in Case 4, OPT can do nothing since no arrivals. The only tricky issue arises in Case 3 since OPT can choose to match the only offline agent with one arrival either from $j = 1$ or $j = 2$. Consider the following two strategies.

The first policy, denoted by **ALG-A**, which aims to maximize $\min(\mathbb{E}[X_1]/\epsilon, \mathbb{E}[X_2]/1)$ on every arrival instance. Thus, in Case 3, **ALG-A** should match the only offline agent with one arrival of $j = 1$ and $j = 2$ with respective probabilities of $\epsilon/(1 + \epsilon)$ and $1/(1 + \epsilon)$. Note that the expected number of matches of $j = 1$ by **ALG-A** should be

$$\mathbb{E}[X_1] = \Pr[\text{Case 2}] \cdot 1 + \Pr[\text{Case 3}] \cdot \frac{\epsilon}{1 + \epsilon} = (1 - e^{-\epsilon}) \cdot (1/e) \cdot 1 + (1 - e^{-\epsilon})(1 - 1/e) \cdot \frac{\epsilon}{1 + \epsilon}.$$

This suggests that

$$\frac{\mathbb{E}[X_1]}{\lambda_1} = \frac{\mathbb{E}[X_1]}{\epsilon} \to \frac{1}{e} \quad (\text{when } \epsilon \to 0).$$

Similarly, the expected number of matches of $j = 2$ by **ALG-A** should be

$$\frac{\mathbb{E}[X_2]}{\lambda_2} = \mathbb{E}[X_2] \to 1 - 1/e \quad (\epsilon \to 0).$$

Thus, we claim that **ALG-A** above achieves a long-run fairness of $\min\left(\frac{\mathbb{E}[X_1]}{\lambda_1}, \frac{\mathbb{E}[X_2]}{\lambda_2}\right) = 1/e$.

Now, we consider a second strategy, denoted by **ALG-B**, which in Case 3 matches the only offline agent with one arrival of $j = 1$ and $j = 2$ with respective probabilities of $p$ and $1 - p$, respectively, with $p \in (0, 1)$ being a constant independent of $\epsilon$. Note that the expected number of matches of $j = 1$ by **ALG-B** should be

$$\mathbb{E}[X_1] = \Pr[\text{Case 2}] \cdot 1 + \Pr[\text{Case 3}] \cdot p = (1 - e^{-\epsilon}) \cdot (1/e) \cdot 1 + (1 - e^{-\epsilon})(1 - 1/e) \cdot p.$$

Thus,

$$\frac{\mathbb{E}[X_1]}{\lambda_1} = \frac{\mathbb{E}[X_1]}{\epsilon} \to \frac{1}{e} + (1 - 1/e) \cdot p \quad (\epsilon \to 0).$$

Similarly, we have

$$\frac{\mathbb{E}[X_2]}{\lambda_2} = \mathbb{E}[X_2] \to 1 - 1/e \quad (\epsilon \to 0).$$

Thus, the long-run fairness achieved by **ALG-B** is equal to

$$\min\left(\frac{\mathbb{E}[X_1]}{\lambda_1}, \frac{\mathbb{E}[X_2]}{\lambda_2}\right) = \min\left(1/e + (1 - 1/e) \cdot p, 1 - 1/e\right),$$

which is equal to $1 - 1/e$ when $p \geq (1 - 2/e)/(1 - 1/e) \approx 0.418$. That is strictly better than **ALG-A**. We can verify that **ALG-B** with $p \geq (1 - 2/e)/(1 - 1/e)$ is a clairvoyant optimal policy.

# B   A Detailed Comparison of Our Work and Ma et al. [28]

Let us detail the differences between the two studies in the following two aspects.

**Model**: As pointed out on lines 155 to 162, the model of [28] assumes integral arrival rates for all online types and then further assumes, without loss of generality, that each online type has a unit arrival rate. In contrast, we do not make that assumption. Importantly, the integral-arrival-rate assumption among online types allows them to propose a significantly stronger benchmark LP than ours. Specifically, the extra constraint $\sum_{j \in S} x_{ij} \leq 1 - e^{-\lambda(S)} = 1 - e^{-|S|}$ in the LP [28] is crucial for overcoming the $1 - 1/e$ barrier for algorithms in [28], where $\lambda(S)$ denotes the total arrival rates among all online types in $S$. As acknowledged there, their algorithms cannot surpass $1 - 1/e$ without that constraint. In our case, while the constraint remains valid, it becomes ineffective compared to its role in [28], which is particularly evident when most online types are rare. For instance, when each online type in $S$ has an arrival rate as small as $1/n^2$, this constraint reduces to $\sum_{j \in S} x_{ij} \leq 1 - e^{-|S|/n^2}$, where the right-hand side approaches one as $n \to \infty$ regardless of the size of $S$.

**Techniques**. Inspired by the insights above, we can no longer exploit any extra constraint to surpass the $1 - 1/e$ barrier. Instead, we conduct a parameter-dependent competitive analysis for our sampling-based policies and explore various scenarios where our algorithm can exceed $1 - 1/e$ or even approach one. Specifically, we incorporate two parameters—the optimal LP value and the minimum offline serving capacity—into the analysis and the final competitive ratio. In contrast, Ma et al. [28] conducted a traditional parameter-free analysis, neglecting the potential impact of different parameters in the input instance on the final competitiveness.

# C   Examples Showing the Possibilities of FAIR-S > FAIR-L **and** FAIR-S < FAIR-L

Below are examples showing it is possible that FAIR-S > FAIR-L and FAIR-S < FAIR-L.

**Example 3.** *Consider a simple example where we have one single offline agent and one single online type with $b = \lambda = 1$. Consider the algorithm FCFS: serve the online agent whenever it arrives.*

*Let $A \sim \mathrm{Pois}(1)$ be the number of arrivals of online agents. Observe that FAIR-L = $\mathbb{E}[X]$ = $\Pr[A \geq 1] = 1 - 1/e$. Note that when $A = 0$, we have FAIR-S = 1. Thus, we can verify that*

$$\text{FAIR-S} = \Pr[A = 0] + \sum_{k=1}^{\infty} \frac{\Pr[A = k]}{k} > e^{-1} + \sum_{k=1}^{\infty} \frac{e^{-1}}{k!} \frac{1}{k+1}$$

$$= e^{-1}\left(1 + \sum_{k=2}^{\infty} \frac{1}{k!}\right) = e^{-1}\left(1 + e - 2\right) = 1 - 1/e = \text{FAIR-L}.$$

Thus we claim that it is possible that FAIR-S > FAIR-L.

**Example 4.** *Consider a simple example where we have one single offline agent and one single online type with $b = 1$ and an online arrival rate of $\lambda$. Consider such an algorithm featured by a threshold $k$ as follows: serve the online agent only when it arrives for the $k$th time. In other words, ignore it for the first $k - 1$ arrivals. Let $A \sim \mathrm{Pois}(\lambda)$ denote the number of online arrivals.*

*Take $\lambda = 10$ and $k = 11$. We can verify that (1) FAIR-L = $\frac{\Pr[A \geq k]}{\lambda}$; (2)*

$$\text{FAIR-S} = \Pr[A = 0] + \sum_{\ell=k}^{\infty} \Pr[A = \ell]/\ell < e^{-\lambda} + \Pr[A \geq k]/k < \Pr[A \geq k]/\lambda = \text{FAIR-L}.$$

*Thus, we claim that it is possible that FAIR-S < FAIR-L.*

## D  Proof of Lemma 4

*Proof.* Part (1) follows from the fact (see *e.g.,* [27]) that $\mathbb{E}\big[\min\{\text{Pois}(b/s), b\}\big]/b$ is increasing in $b$. Part (2) is valid since: if $s \leq 1$, then $\kappa(b,s) = \mathbb{E}\big[\min\{\text{Pois}(b/s), b\}\big]/b$ which is decreasing in $s$; if $s \geq 1$, then $\kappa(b,s) = s \cdot (\mathbb{E}\big[\min\{\text{Pois}(b/s), b\}\big]/b)$, which is increasing in $s$. Furthermore, we can derive that

$$\frac{\mathbb{E}[\min\{\text{Pois}(b/s), b\}]}{b} = 1 - \frac{1}{b}\mathbb{E}[\max\{b - \text{Pois}(b/s), 0\}] = 1 - \sum_{k=0}^{b-1} e^{-b/s}\frac{b^{k-1}}{s^k k!}(b-k);$$

if $s = 1$ then this equals

$$\frac{\mathbb{E}[\min\{\text{Pois}(b/s), b\}]}{b} = 1 - \sum_{k=0}^{b-1} e^{-b}\frac{b^k}{k!} + \sum_{k=1}^{b-1} e^{-b}\frac{b^{k-1}}{(k-1)!} = 1 - e^{-b}\frac{b^{b-1}}{(b-1)!} = \kappa(b,1).$$

It can be verified that $\kappa(b,1)$ gets minimized at $b = 1$ with $\kappa(1,1) = 1 - 1/e$. For $b > 1$,

$$\kappa(b,1) \geq 1 - e^{-b}\frac{b^{b-1}}{\frac{(b-1)^{b-1}}{e^{b-1}}\sqrt{2\pi(b-1)}} = 1 - \frac{1}{e}(1 + \frac{1}{b-1})^{b-1}\frac{1}{\sqrt{2\pi(b-1)}} \geq 1 - \frac{1}{\sqrt{2\pi(b-1)}},$$

where we use Stirling's approximation in the first inequality. This establishes Part (3) and Part (5).

Now, we show Parts (4) and (6). Recall that $\eta(\lambda, s) = \frac{\mathbb{E}[\min \text{Pois}(\lambda), \lambda s]}{\lambda \cdot \min(s,1)}$ and $\kappa(b,s) = \eta(b/s, s)$. Consider the first case when $s > 1$. We see that

$$\eta(\lambda, s) = \frac{\mathbb{E}[\min(\text{Pois}(\lambda), \lambda s)]}{\lambda} \geq \frac{1}{\lambda}\sum_{k=1}^{\lambda s}\frac{e^{-\lambda}\lambda^k k}{k!} = \sum_{k=0}^{\lambda s - 1}\frac{e^{-\lambda}\lambda^k}{k!} = 1 - \Pr[\text{Pois}(\lambda) \geq \lambda s]$$

$$\geq 1 - \exp\left(-\lambda \cdot \frac{\ln s \cdot (s-1)^2}{s} \cdot (1 - o(1))\right).$$

The last inequality is due to the upper tail bound of a Poisson random variable as shown by [8], where $o(1) = \Theta(1/\ln s)$ is vanishing when $s \to \infty$. Thus, since $\kappa(b,s) = \eta(b/s, s)$, we see $\kappa(b,s) \geq 1 - \exp(-b \cdot \ln s \cdot (1 - 1/s)^2(1 - o(1)))$, completing Part (4).

Similarly, for $s < 1$, we have

$$\eta(\lambda, s) = \frac{\mathbb{E}[\min(\text{Pois}(\lambda), \lambda s)]}{\lambda s} \geq \frac{\lambda s}{\lambda s}\sum_{k=\lambda s}^{\infty}\frac{e^{-\lambda}\lambda^k}{k!} = 1 - \Pr[\text{Pois}(\lambda) < \lambda s]$$

$$\geq 1 - \exp\left(-\frac{\lambda(1-s)^2}{2}\right).$$

The last inequality is due to [8]. Thus, by replacing $\lambda$ with $b/s$, we establish Part (6). $\qquad\square$

## E  An Alternative Proof for a Stronger Version of Lemma 5

Consider Example 2. We present an alternative proof for a stronger version of Lemma 5 below.

**Lemma 6.** *For Example 1, the optimal non-rejecting policy (*Non-Rej*) achieves a long-run fairness of $1/2 + o(1)$ and $1 - o(1)$ for the rare type and the common type, respectively, where $o(1)$ is a vanishing term when $n \to \infty$.*

Since the clairvoyant optimal achieves a long-run fairness of $1 - O(1/n)$ for both common and rare types, the above lemma immediately implies that Non-Rej achieves a competitiveness of $1/2 + o(1)$.

We claim that Non-Rej above is an optimal non-rejecting algorithm for the bad example. This can be justified as follows: When a common type arrives, the best strategy is to sample an available offline neighbor (server) uniformly at random since each rare type has an equal chance of arriving subsequently.

---

**ALGORITHM 2:** An Optimal Non-Rejecting Policy for Example 2 (Non-Rej)

---

**1** Let an online agent $j$ arrive at time $t$.
**2** **if** *$j$ is of the common type* **then**
**3**    |   Sample an available neighbor uniformly at random, if any, and assign $j$ to it;
**4** **else**
**5**    |   Assign $j$ to its unique offline neighbor, if it is available then.

---

*Proof of Lemma 6.* For each $k$ with $0 \leq k \leq n$ and $t \in [0, 1]$, let $\pi_k(t)$ be the probability that there are $k$ available servers at time $t$. By the nature of Non-Rej, the update on the number of available servers follows a pure death process: The system starts at state $k = n$ at time $t = 0$, i.e., $\pi_n(0) = 1$; given the system has $1 \leq k < n$ free servers at time $t$, it leaves the state whenever either a common type arrives or any rare type uniquely served by any of the $k$ free servers arrives, with a total arrival rate of $n - 1 + k/n$. Thus,

$$\frac{\mathsf{d}\pi_k(t)}{\mathsf{d}t} = -\pi_k(t) \cdot n, \pi_k(0) = 1 \qquad\qquad\qquad\qquad k = n;$$

$$\frac{\mathsf{d}\pi_k(t)}{\mathsf{d}t} = -\pi_k(t) \cdot \left(n - 1 + \frac{k}{n}\right) + \pi_{k+1}(t)\left(n - 1 + \frac{k+1}{n}\right), \pi_k(0) = 0 \qquad 1 \leq k < n;$$

$$\pi_0(t) = 1 - \sum_{k=1}^{n} \pi_k(t).$$

From the above ordinary-differential-equation system, we can solve that

$$\pi_k(t) = \begin{cases} \Pr[\mathrm{Pois}(tn) = n - k](1 + o(1)), & \text{if } 1 \leq k \leq n; \\ \Pr[\mathrm{Pois}(tn) \geq n](1 + o(1)), & \text{if } k = 0. \end{cases}$$

Let $X$ be the (random) numbers of rare type agents that arrive and get served in Non-Rej. Therefore,

$$\mathbb{E}[X] = \int_0^1 \sum_{k=1}^{n} \pi_k(t) \cdot (k/n)\mathsf{d}t = \int_0^1 \sum_{k=1}^{n} \Big( \Pr[\mathrm{Pois}(tn) = n - k](1 + o(1)) \Big) \cdot (k/n)\mathsf{d}t$$

We can verify that $\mathbb{E}[X] = 1/2 + o(1)$, where $o(1)$ vanishes as $n \to \infty$. By symmetry, the number of each rare type served equals $\mathbb{E}[X]/n$, leading to a long-run fairness of $(\mathbb{E}[X]/n)/(1/n) = \mathbb{E}[X] = 1/2 + o(1)$. Similarly, let $Y$ be the number of common type agents that arrive and are served in Non-Rej. Thus, the long-run fairness achieved for the common type is

$$\frac{\mathbb{E}[Y]}{n-1} = \frac{1}{n-1} \int_0^1 \sum_{k=1}^{n} \pi_k(t) \cdot (n-1)\mathsf{d}t = \int_0^1 \sum_{k=1}^{n} \Big( \Pr[\mathrm{Pois}(tn) = n - k](1 + o(1)) \Big)\mathsf{d}t$$

$$= \int_0^1 \Pr[\mathrm{Pois}(tn) \leq n - 1]\mathsf{d}t + o(1) = \frac{1}{n} \int_0^1 \Pr[\mathrm{Pois}(tn) \leq n - 1] \cdot n \, \mathsf{d}t + o(1)$$

$$= \frac{1}{n} \cdot \mathbb{E}\Big[ \min\Big(\mathrm{Pois}(n), n\Big) \Big] - o(1) = 1 - o(1).$$

Thus, we establish the claim. $\qquad\qquad\qquad\qquad\qquad\qquad\qquad\qquad\qquad\qquad\qquad\qquad\qquad\qquad$ □

## F    A Quantitative Study on the Tradeoff between Competitiveness and the Number of Rejected Agents

We use the bad example (Example 2) to conduct a case study on the trade-off between competitiveness and the number of rejected arriving agents (when the serving capacity remains). We hope this case study can provide insights for a comprehensive study of this trade-off in general cases. The proof of Lemma 6 in Section E offers a foundational framework for analyzing a general class of online policies. Consider an updated version of Non-Rej, denoted by Rej($\alpha$), where we reject each arriving common-type agent with a preset constant probability $\alpha \in [0, 1]$. We expect the parameter $\alpha$ to serve as a useful moderator in balancing competitiveness and the number of rejected common-type agents.

**ALGORITHM 3:** An Updated Policy Parameterized by $\alpha \in [0, 1]$: $\mathsf{Rej}(\alpha)$

---

**1** Let an online agent $j$ arrive at time $t$.

**2** **if** *$j$ is of the common type* **then**

**3** $\quad$ With probability $1 - \alpha$, reject $j$; With probability $\alpha$, sample an available neighbor uniformly at random, if any, and assign $j$ to it;

**4** **else**

**5** $\quad$ Assign $j$ to its unique offline neighbor, if it is available then.

---

Note that: (1) When $\alpha = 1$, $\mathsf{Rej}(\alpha)$ reduces to Non-Rej; (2) The expected number of arriving common-type agents rejected in $\mathsf{Rej}(\alpha)$ is at least $(1 - \alpha)(n - 1)$.

**Lemma 7.** *The policy $\mathsf{Rej}(\alpha)$ achieves a long-run fairness of $1 - \alpha/2 + o(1)$ and $\alpha - o(1)$ for the rare and common types, respectively, where $o(1)$ is a vanishing term as $n \to \infty$. Additionally, it rejects at least $(1 - \alpha)(n - 1)$ arriving common-type agents in expectation.*

*Proof.* For each $k$ with $0 \leq k \leq n$ and $t \in [0, 1]$, let $\pi_k(t)$ be the probability that there are $k$ available servers at time $t$. Following the same analysis as in the proof of Lemma 6, we have

$$\pi_k(t) = \begin{cases} \Pr[\mathrm{Pois}(tn\alpha) = n - k](1 + o(1)), & \text{if } 1 \leq k \leq n; \\ \Pr[\mathrm{Pois}(tn\alpha) \geq n](1 + o(1)), & \text{if } k = 0. \end{cases}$$

Let $X$ be the (random) numbers of rare type agents that arrive and get served in $\mathsf{Rej}(\alpha)$. Therefore,

$$\mathbb{E}[X] = \int_0^1 \sum_{k=1}^n \pi_k(t) \cdot (k/n) \mathrm{d}t = \int_0^1 \sum_{k=1}^n \Big( \Pr[\mathrm{Pois}(tn\alpha) = n - k](1 + o(1)) \Big) \cdot (k/n) \mathrm{d}t$$

$$= \int_0^1 \sum_{\ell=0}^{n-1} \Pr[\mathrm{Pois}(tn\alpha) = \ell] \cdot (1 - \ell/n) \mathrm{d}t + o(1)$$

$$= \int_0^1 \sum_{\ell=0}^{n-1} \Pr[\mathrm{Pois}(tn\alpha) = \ell] \mathrm{d}t - \frac{\ell}{n} \int_0^1 \sum_{\ell=0}^{n-1} \Pr[\mathrm{Pois}(tn\alpha) = \ell] \mathrm{d}t + o(1)$$

$$= \int_0^1 \Pr[\mathrm{Pois}(tn\alpha) \leq n - 1] \mathrm{d}t - \int_0^1 (t\alpha) \cdot \Pr[\mathrm{Pois}(tn\alpha) \leq n - 2] \mathrm{d}t + o(1)$$

$$= 1 - \alpha/2 + o(1).$$

By symmetry, the number of each rare type served is equal to $\mathbb{E}[X]/n$, leading to a long-run fairness of $(\mathbb{E}[X]/n)/(1/n) = \mathbb{E}[X] = 1 - \alpha/2 + o(1)$. Similarly, let $Y$ be the number of common type agents that arrive and are served in $\mathsf{Rej}(\alpha)$. The long-run fairness achieved for the common type is

$$\frac{\mathbb{E}[Y]}{n - 1} = \frac{1}{n - 1} \int_0^1 \sum_{k=1}^n \pi_k(t) \cdot (n - 1) \cdot \alpha \, \mathrm{d}t = \alpha \int_0^1 \sum_{k=1}^n \Pr[\mathrm{Pois}(tn\alpha) = n - k] \mathrm{d}t + o(1)$$

$$= \alpha \int_0^1 \Pr[\mathrm{Pois}(tn\alpha) \leq n - 1] \mathrm{d}t + o(1) = \alpha \cdot \frac{1}{n\alpha} \int_0^1 \Pr[\mathrm{Pois}(tn\alpha) \leq n - 1] \cdot (n\alpha) \, \mathrm{d}t + o(1)$$

$$= \alpha \cdot \frac{1}{n\alpha} \cdot \mathbb{E}\Big[ \min \Big( \mathrm{Pois}(n\alpha), n \Big) \Big] - o(1) = \alpha - o(1).$$

Thus, we establish the claim. $\qquad\qquad\qquad\qquad\qquad\qquad\qquad\qquad\qquad\qquad\qquad\square$

## G    Proof of Theorem 3: Extension of OM-LF to Group-Level Fairness

In this section, we consider the extension of OM-LF to group-level fairness. Recall that we have a collection of protected groups $\mathcal{G} = \{g | g \subseteq J\}$, where each group $g \in \mathcal{G}$ is a subset of $J$ that indicates the online agent types in group $g$. The updated long-run fairness with respect to $\mathcal{G}$ is defined

as FAIR-L$(\mathcal{G}) = \min_{g \in \mathcal{G}} \frac{\mathbb{E}_{\mathcal{A},\text{ALG}}[X(g)]}{\sum_{j \in g} \lambda_j}$, as shown in (6). Below is an updated version of Benchmark LP for the long-run group-level fairness.

$$\max \ s \tag{10}$$

$$\sum_{j \in \mathcal{N}_i} x_{ij} \leq b_i \qquad \forall i \in I$$

$$\sum_{j \in g} \sum_{i \in \mathcal{N}_j} x_{ij} \geq s \cdot \sum_{j \in g} \lambda_j \ \ \forall g \in \mathcal{G} \tag{11}$$

$$\sum_{i \in \mathcal{N}_j} x_{ij} \leq \lambda_j \qquad \forall j \in J \tag{12}$$

$$s, x_{ij} \geq 0 \qquad \forall (i,j) \in E$$

Note that we add a new set of constraints (12), which are clearly valid for any clairvoyant algorithm since the constraints hold on every sample path based on the realized number of arrivals and services. Therefore, OPT $\leq s^*$, where $s^*$ represents the optimal value of LP (10), following the same argument as presented in the proof of Lemma 1.

### G.1 Proof of Part (1) of Theorem 3: SAMP-G and Competitive Analysis

The algorithm SAMP-G for OM-LF under long-run group-level fairness is formally stated in Algorithm 4. Note that SAMP-G will reject an online agent immediately with probability $1 - \sum_{i \in \mathcal{N}_j} x_{ij}^*/\lambda_j$, and will also reject it if the first sampled offline agent has reached capacity.

---

**ALGORITHM 4:** A Sampling Algorithm for OM-LF under Long-Run Group-Level Fairness (SAMP-G)

---

1 Solve LP (10) to get an optimal solution $\{x_{ij}^*\}$.
2 Let an online agent (of type) $j$ arrive at time $t$.
3 Sample a neighbor $i \in \mathcal{N}_j$ with probability $x_{ij}^*/\lambda_j$.
   /* This is a valid distribution due to Constraint (12).                              */
4 If $i$ is safe, *i.e.,* $i$ has remaining capacity, then assign $i$ to serve $j$; otherwise, reject $j$.

---

*Proof of Part (1) of Theorem 3.* We provide a terse argument since detailed logic can be found in Lemmas 2–3. The incoming demand flow to an offline agent $i \in I$ is Poisson with rate $\sum_{j \in \mathcal{N}_i} \lambda_j \frac{x_{ij}^*}{\lambda_j}$, which is at most $b_i$ by LP feasibility. Therefore, the capacity of any offline agent $i$ has not been reached at time $t$ with probability at least $\Pr[\text{Pois}(b_i t) < b_i]$. Using this fact, the expected number of times offline agent $i$ serves online type $j$ is at least

$$\int_0^1 \lambda_j \frac{x_{ij}^*}{\lambda_j} \Pr[\text{Pois}(b_i t) < b_i] dt = \frac{x_{ij}^*}{b_i} \cdot \int_0^1 b_i \cdot \Pr\left[\text{Pois}(b_i t) < b_i\right] dt$$

$$= x_{ij}^* \cdot \frac{\mathbb{E}[\min\{\text{Pois}(b_i), b_i\}]}{b_i}.$$

Applying Lemma 4 twice, the expected total number of times a group $g \in \mathcal{G}$ is served is at least

$$\sum_{j \in g} \sum_{i \in \mathcal{N}_j} x_{ij}^* \cdot \frac{\mathbb{E}[\min\{\text{Pois}(b_i), b_i\}]}{b_i} \geq \frac{\mathbb{E}[\min\{\text{Pois}(b), b\}]}{b} \cdot \sum_{j \in g} \sum_{i \in \mathcal{N}_j} x_{ij}^*$$

$$= (1 - e^{-b} \frac{b^b}{b!}) \cdot \sum_{j \in g} \sum_{i \in \mathcal{N}_j} x_{ij}^*.$$

The proof is completed by using the LP inequality that $\sum_{j \in g} \sum_{i \in \mathcal{N}_j} x_{ij}^* / \sum_{j \in g} \lambda_j \geq s^*$. $\qquad\square$

**Remarks on the Missing Role Played by $s^*$ in the Final Competitiveness.** Below are a few notes on why the competitiveness should no longer depend on $s^*$. Recall that in Theorem 1, when the focus

is the fairness among all possible online types, $s^*$ was interpreted as the "scale" of demand that can be served, and the competitiveness approached 1 if $s^* \to \infty$ or $s^* \to 0^+$. However, in the context of group-level fairness, $s^*$ no longer has this interpretation, and the statements about asymptotic optimality no longer hold. To illustrate this, we provide two examples below.

First, $s^* \to \infty$ is no longer possible, because $s^* \leq 1$ is implied by Constraints (11) and (12). On the other hand, if we do not add these constraints, then the LP has an unbounded gap, as demonstrated by the following example. There is a single group consisting of $n$ types with arrival rates 1. One type is connected to an offline agent with capacity $n$; the other types are connected to no offline agents. Without constraints (12), the LP would be able to "overserve" the first type and achieve a fairness of 1; any actual algorithm would have a fairness at most $1/n$. All in all, in the generalized model, it is no longer possible to allow an $s$ which is greater than 1.

Second, if $s^* \to 0^+$, it is no longer the case that online algorithms can achieve a fairness of $s^*$, as demonstrated by the following example. There is a single group consisting of 2 types; one with arrival rate 1 and the other with arrival rate $\lambda \to \infty$. Each type is connected to its own offline agent with capacity 1. In this case $s^* = 2/(1+\lambda)$, which approaches 0. However, an online algorithm makes in expectation only $1 + (1 - 1/\mathrm{e})$ services, achieving fairness $(2 - 1/\mathrm{e})/(1 + \lambda)$.

## G.2 Algorithm RESERVE when All Online Types are Common

In this section we introduce another regime, in which algorithms are 1-competitive—the regime where all online types are common, *i.e.,* all have high arrival rates. However, this regime requires a different algorithm, which we now motivate using the following example.

**Example 5.** *$J$ consists of a single type $a$ with $\lambda_a = n$ and $I$ consists of $n$ separate servers each with unit capacity. Using SAMP-G, each server faces a separate demand according to a Poisson process of rate 1 and successfully serves demand with probability $1 - 1/\mathrm{e}$. The total expected demand served is $n(1 - 1/\mathrm{e})$. However, an algorithm that adaptively chooses an available server and never rejects incoming demand as long as a server is available serves a total expected demand of $\mathbb{E}[\min\{\mathrm{Pois}(n), n\}]$. As $n \to \infty$, the FAIR-L of the adaptive algorithm approaches 1, while the FAIR-L of SAMP-G is stuck at $1 - 1/\mathrm{e}$.*

SAMP-G did not improve on this example even when the arrival rate approached $\infty$ because it did not "pool" the servers in order to reduce the variance in demand served. Motivated by this example, we now introduce an algorithm RESERVE, which pre-assigns the capacity that will be used to serve each online type. In general, offline agents could be adjacent to many online types and may not be as straightforward to assign as in Example 5; however, we make use of the updated LP (10) along with the dependent rounding procedure [16] to generate a randomized assignment. We state RESERVE in Algorithm 5 and leave the proof of Part (2) of Theorem 3 to Appendix.

---

**ALGORITHM 5:** Alternate Algorithm that Pre-reserves Capacities (RESERVE)

---
1 Split and re-index offline agents as necessary so that $b_i = 1$ for all $i \in I$.
2 Solve LP (10) to get an optimal solution $\{x_{ij}^*, s^*\}$, and define $x_j^* = \sum_{i \in \mathcal{N}_j} x_{ij}^*$ for all $j \in J$. Note that $x_j^* \leq \lambda_j$ for all $j \in J$, by Constraint (12).
3 Apply dependent rounding [16] to the LP solution $\{x_{ij}^*\}$, and let $\{X_{ij}^R\}$ be the rounded binary vector such that $\sum_{j \in \mathcal{N}_i} X_{ij}^R \leq 1$ for all $i \in I$.
4 For each online type $j$, reserve the offline agents $\{i : X_{ij}^R = 1\}$ exclusively for serving $j$, and match them to incoming type-$j$ agents in a first-come-first-serve manner.

---

## G.3 Proof of Part (2) of Theorem 3

*Proof.* For all $j$, let $\mathrm{Serve}(j)$ denote the set $\{i : X_{ij}^R = 1\}$, which is generally randomized. By the work of [16], it is possible to do the rounding in Step 3 so that the sets $\{\mathrm{Serve}(j) : j \in J\}$ are always mutually disjoint, and $|\mathrm{Serve}(j)| \in \{\lfloor x_j^* \rfloor, \lceil x_j^* \rceil\}$ for all $j$ with $\mathbb{E}[|\mathrm{Serve}(j)|] = x_j^*$. $\mathrm{Serve}(j)$ is fixed in advance, and hence independent of the number of arrivals of type $j$, for any $j \in J$. Therefore,

the expected number of an online type $j$ served is

$$\mathbb{E}[\min\{\mathrm{Pois}(\lambda_j), |\mathrm{Serve}(j)|\}]$$
$$= \mathbb{E}[\mathrm{Pois}(\lambda_j) \cdot \mathbf{1}(\mathrm{Pois}(\lambda_j) \le \lfloor x_j^* \rfloor) + |\mathrm{Serve}(j)| \cdot \mathbf{1}(\mathrm{Pois}(\lambda_j) > \lfloor x_j^* \rfloor)]$$
$$= \mathbb{E}[\mathrm{Pois}(\lambda_j) \cdot \mathbf{1}(\mathrm{Pois}(\lambda_j) \le \lfloor x_j^* \rfloor)] + \mathbb{E}[|\mathrm{Serve}(j)|] \Pr[\mathrm{Pois}(\lambda_j) > \lfloor x_j^* \rfloor]$$
$$= \mathbb{E}[\mathrm{Pois}(\lambda_j) \cdot \mathbf{1}(\mathrm{Pois}(\lambda_j) \le \lfloor x_j^* \rfloor)] + x_j^* \Pr[\mathrm{Pois}(\lambda_j) > \lfloor x_j^* \rfloor]$$
$$= \mathbb{E}[\min\{\mathrm{Pois}(\lambda_j), x_j^*\}],$$

where the first equality uses the property that $|\mathrm{Serve}(j)| \in \{\lfloor x_j^* \rfloor, \lceil x_j^* \rceil\}$, the second equality uses independence, and the third equality uses the property that $\mathbb{E}[|\mathrm{Serve}(j)|] = x_j^*$.

For any group $g \in \mathcal{G}$, the expected fraction served is

$$\frac{\sum_{j \in g} \mathbb{E}[\min\{\mathrm{Pois}(\lambda_j), x_j^*\}]}{\sum_{j \in g} \lambda_j} \ge \frac{\sum_{j \in g} \frac{\mathbb{E}[\min\{\mathrm{Pois}(\lambda_j), \lambda_j\}]}{\lambda_j} \cdot x_j^*}{\sum_{j \in g} \lambda_j}$$
$$\ge \frac{\mathbb{E}[\min\{\mathrm{Pois}(\lambda), \lambda\}]}{\lambda} \cdot \frac{\sum_{j \in g} x_j^*}{\sum_{j \in g} \lambda_j} = \left(1 - e^{-\lambda} \frac{\lambda^\lambda}{\lambda!}\right) \cdot \frac{\sum_{j \in g} x_j^*}{\sum_{j \in g} \lambda_j},$$

where the first inequality holds because $1 \ge \frac{x_j^*}{\lambda_j}$, and the second inequality holds because $\frac{\mathbb{E}[\min\{\mathrm{Pois}(\lambda), \lambda\}]}{\lambda}$ is increasing in $\lambda$. Finally, $\frac{\sum_{j \in g} x_j^*}{\sum_{j \in g} \lambda_j} \ge s^*$ by LP feasibility, where $s^*$ is in turn an upper bound on OPT. Since this holds for all groups $g \in \mathcal{G}$, the proof is complete. $\qquad \square$

## H  Proof of Theorem 4: Short-Run Fairness with a Single Offline Type

### H.1  Proof of Part (1) of Theorem 4: Competitive Analysis for FCFS

*Proof.* Let $b$ be the serving capacity of the single offline agent and $\lambda$ be the total arrival rate of online types. Thus, the fairness of FCFS under FAIR-S should be at least $\Pr[\mathrm{Pois}(\lambda) \le b]$. In contrast, the fairness of the offline optimal under FAIR-S should be OPT $= \Pr[\mathrm{Pois}(\lambda) \le b] + \sum_{k>b} \Pr[\mathrm{Pois}(\lambda) = k] \cdot (b/k)$. By definition, the competitive ratio of FCFS under FAIR-S is at least

$$f(b, \lambda) \doteq \frac{\Pr[\mathrm{Pois}(\lambda) \le b]}{\Pr[\mathrm{Pois}(\lambda) \le b] + \sum_{k>b} \Pr[\mathrm{Pois}(\lambda) = k] b/k}.$$

We now show that the value of $f(b, \lambda)$, for all positive integers $b$ and $\lambda \le 1$, is lower-bounded by $f(1, 1)$, which equals approximately 0.863. We first show that for any given $\lambda \le 1$, $f(b, \lambda)$ is an increasing function of $b$ when $b \ge 1$. Fix a $\lambda \in [0, 1]$, Let $f_1(b) = \Pr[\mathrm{Pois}(\lambda) \le b]$ and $f_2(b) = \Pr[\mathrm{Pois}(\lambda) \le b] + \sum_{k>b} \Pr[\mathrm{Pois}(\lambda) = k] b/k$. Thus, we have $f(b, \lambda) = f_1(b,)/f_2(b)$. Observe that (1) $f_1(b) - f_1(b-1) = \Pr[\mathrm{Pois}(\lambda) = b] = e^{-\lambda} \lambda^b / b!$; (2) $f_2(b) - f_2(b-1) = \sum_{k=b}^{\infty} e^{-\lambda} \lambda^k / (k \cdot k!)$. Thus, for $b \ge 2$,

$$\frac{f_1(b) - f_1(b-1)}{f_2(b) - f_2(b-1)} = \frac{e^{-\lambda} \lambda^b / b!}{\sum_{k=b}^{\infty} e^{-\lambda} \lambda^k / (k \cdot k!)} \ge \frac{1}{\sum_{k=b}^{\infty} b! / (k \cdot k!)} \ge \frac{1}{\sum_{k=2}^{\infty} 2! / (k \cdot k!)} = 1.573.$$

Note that $f_1(b)/f_2(b) \le 1$. Thus, we claim that $f(b, \lambda) = f_1(b)/f_2(b)$ is an increasing function of $b \ge 1$. So, $f(b, \lambda) \ge f(1, \lambda)$.

Now we show $f(1, \lambda)$ is a decreasing function of $\lambda \in [0, 1]$. When $b = 1$, we have

$$f(1, \lambda) = \frac{e^{-\lambda}(1 + \lambda)}{e^{-\lambda}(1 + \lambda) + \sum_{k=2}^{\infty} \frac{e^{-\lambda} \lambda^k}{k! k}} = \frac{1}{1 + \frac{\lambda^k}{1+\lambda} \sum_{k=2}^{\infty} \frac{1}{k! k}}.$$

Observe that $\lambda^k/(1 + \lambda)$ increases over $\lambda > 0$ for all given integer $k \ge 1$. Thus, we claim $f(1, \lambda)$ is a decreasing function of $\lambda$ over $\lambda \in [0, 1]$. Therefore, $f(1, \lambda) \ge f(1, 1) \approx 0.863$. $\qquad \square$

## H.2 Proof of Part (2) of Theorem 4

*Proof.* Let $b$ be the serving capacity of the single offline agent and $\lambda$ be the total arrival rate of online types. Consider an instance with $b = 1$, and assume all online types are rare. In other words, with probability one, every online type has at most one arrival. For each $t \in [0, 1]$, let $\sigma(\lambda, t)$ be the fairness achieved by an optimal online algorithm under FAIR-S when the online process is restricted as Poisson process of rate $\lambda t$. Thus, we care about the value $\sigma(\lambda, 1)$, which is the fairness achieved by the online optimal.

Consider an infinitesimally small period $\delta$ during which at most one arrival can occur. Now we try to upper bound $\sigma(\lambda, t + \delta)$. (Case 1) There is no arrival during $(t, t + \delta]$ which occurs with probability $e^{-\lambda\delta}$. In the case, we have $\sigma(\lambda, t + \delta) = \sigma(\lambda, t)$. (Case 2) There is one arrival during $(t, t + \delta]$ which occurs with probability $1 - e^{-\lambda\delta}$. In this case, we have $\sigma(\lambda, t+\delta) \leq \min(\sigma(\lambda, t), 1 - \sigma(\lambda, t) + e^{-\lambda t})$, which is shown as below.

Let $\alpha_{t,k}$ be the fairness achieved by an online optimal when there are $k$ arrivals during $[0, t]$. Observe that $\alpha_{t,0} = 1$ for all $t \in [0, 1]$. Therefore, by definition, $\sigma(\lambda, t) = \sum_{k=0}^{\infty} \alpha_{t,k} \Pr[\mathrm{Pois}(\lambda t) = k]$. Assume there is one arrival during $(t, t + \delta]$. Note that

$$\sigma(\lambda, t + \delta) = \sum_{k=0}^{\infty} \min(\alpha_{t,k}, 1 - k \cdot \alpha_{t,k}) \Pr[\mathrm{Pois}(\lambda t) = k]$$

$$\leq \sum_{k=0}^{\infty} \alpha_{t,k} \Pr[\mathrm{Pois}(\lambda t) = k] = \sigma(\lambda, t),$$

$$\sigma(\lambda, t + \delta) \leq \sum_{k=0}^{\infty} (1 - k \cdot \alpha_{t,k}) \Pr[\mathrm{Pois}(\lambda t) = k]$$

$$\leq 1 - \sum_{k=1}^{\infty} \alpha_{t,k} \Pr[\mathrm{Pois}(\lambda t) = k] = 1 - (\sigma(\lambda, t) - e^{-\lambda t}).$$

Thus, we claim that $\sigma(\lambda, t + \delta) \leq \min(\sigma(\lambda, t), 1 - \sigma(\lambda, t) + e^{-\lambda t})$. Wrapping up all the above analysis, we have $\sigma(\lambda, t + \delta) \leq e^{-\lambda\delta}\sigma(\lambda, t) + (1 - e^{-\lambda\delta}) \min(\sigma(\lambda, t), 1 - \sigma(\lambda, t) + e^{-\lambda t})$. This suggests that $\partial\sigma(\lambda, t)/\partial t \leq -\lambda\sigma(\lambda, t) + \lambda \min(\sigma(\lambda, t), 1 - \sigma(\lambda, t) + e^{-\lambda t})$.

For each given $\lambda \in [0, 1]$, let $R_\lambda(t)$ be the unique function satisfying that $dR_\lambda(t)/dt = -\lambda R_\lambda(t) + \lambda \min(R_\lambda(t), 1 - R_\lambda(t) + e^{-\lambda t})$ with $R_\lambda(0) = 1$. Thus, we claim that $\sigma(\lambda, 1) \leq R_\lambda(1)$. Recall that the offline optimal has a performance of $e^{-\lambda}(1 + \lambda) + \sum_{k=2}^{\infty} \frac{e^{-\lambda}\lambda^k}{k!k}$ under FAIR-S. We can numerically verify that $R_\lambda(1)/\left(e^{-\lambda}(1 + \lambda) + \sum_{k=2}^{\infty} \frac{e^{-\lambda}\lambda^k}{k!k}\right)$ gets its minimum value of $0.942$ when $\lambda = 1$. Thus, we establish our result. □

## H.3 Proof of Part (3) of Theorem 4: Asymptotically Optimal Algorithhm of Prob-Rej for FAIR-S with a Single Offline Agent

Let $\mathcal{I}(b, \lambda)$ denote an instance of online matching under short-run fairness FAIR-S with a single offline agent of capacity $b$ and a total online arrival rate of $\lambda \gg 1$.

**Lemma 8.** *Consider an instance $\mathcal{I}(b, \lambda)$ with $b/\lambda < 1$. We have* $\mathsf{OPT} \leq (b/\lambda) \cdot \left(1 + 1/\lambda + o(1/\lambda)\right)$.

*Proof.* Let $A = \sum_{j \in J} A_j$ be the total number of online arrivals. Observe that the performance of an optimal clairvoyant under FAIR-S should satisfy (1) $\mathsf{OPT}(A) = 1$ when $A \leq b$ and (2) $\mathsf{OPT}(A) = b/k$ when $A = k > b$. Therefore,

$$\mathsf{OPT} = \mathbb{E}_A[\mathsf{OPT}(A)]$$

$$= \Pr[A \leq b] \cdot 1 + \sum_{k>b}^{\infty} \Pr[A = k] \cdot \frac{b}{k}$$

$$\leq \Pr[A \leq \lambda(1 - (1 - \kappa))] + b \cdot \sum_{k=1}^{\infty} \frac{e^{-\lambda}\lambda^k}{k!} \frac{1}{k}$$

$$\leq \exp\left(-\frac{\lambda(1-\kappa)^2}{2}\right) + b \cdot \left(\frac{1}{\lambda} + \frac{1}{\lambda^2} + o\left(\frac{1}{\lambda^2}\right)\right) = \kappa\left(1 + \frac{1}{\lambda} + o\left(\frac{1}{\lambda}\right)\right).$$

Note that the inequality on the last line is due to the lower tail bound of a Poisson random variable as shown by [8]. Another trick involved is $\sum_{k=1}^{\infty} \frac{\lambda^k}{k!} \frac{1}{k} = \mathrm{Ei}(\lambda) - \ln\lambda - \gamma$ where $\gamma \doteq \lim_{n\to\infty}\left(\sum_{k=1}^{n} 1/k - \ln n\right) \approx 0.577$ is a constant, and Ei is the Exponential integral function. As shown by [32], $\mathrm{Ei}(\lambda) = (e^\lambda/\lambda)(1 + 1/\lambda + o(1/\lambda))$ when $\lambda \gg 1$. Thus, we are done. $\qquad\square$

Now, we formally present the algorithm Prob-Rej in Algorithm 6, which shares the spirit as RESERVE as shown in Section G.2. Part (3) of Theorem 4 shows that Prob-Rej is 1-competitive as the total arrival rate $\lambda \to \infty$, even if the service capacity $b$ is increasing at the same time. Depending on whether $b/\lambda$ is greater than 1, the probabilistic rejection probabilities have to be chosen differently. Also, note that due to dependent rounding, Prob-Rej is less likely to reject an agent if other agents have already been rejected, distributing equal opportunity among the first $K$ arrivals to be served. This dependent rounding makes it different from SAMP, SAMP-G, and similar algorithms in the literature.

---

**ALGORITHM 6:** Probabilistic-Rejection Algorithm for FAIR-S with a Single Offline Agent (Prob-Rej)

---

1 Set $\epsilon = b/\lambda - 1$ if $b/\lambda > 1$ and $\epsilon = \sqrt{\ln\lambda/\lambda}$ otherwise. let $K = \lfloor\lambda(1+\epsilon)\rfloor$.
2 Apply dependent rounding [16] to the vector $\mathbf{x} = (b/K) \cdot \mathbf{1}$, which has $K$ identical entries each equal $b/K$.
   Let $(Y_k)_k \in \{0,1\}^K$ be the random vector output.
3 Suppose an online agent (of type) $j$ arrives, and let it be the $k$th arrival among all online arrivals.
4 If $k \leq K$ and $Y_k = 1$, then serve the incoming type-$j$ agent if it is possible; otherwise, reject agent $j$.

---

*Proof of Part (3) of Theorem 4.* For notation convenience, we use $\mathcal{I}$ to denote $\mathcal{I}(b,\lambda)$, which represents an instance under FAIR-S with a single offline agent of capacity $b$ and a total online arrival rate of $\lambda$. Additionally, we use ALG to refer to the probabilistic-rejection algorithm Prob-Rej. By definition, we have $\mathsf{ALG}(\mathcal{I}) = \mathbb{E}_{\mathcal{A}}\left[\min_{j:A_j>0}\mathbb{E}_{\mathsf{ALG}}[X_j]/A_j\right] \doteq \mathbb{E}_{\mathcal{A}}[\mathsf{ALG}(\mathcal{A})]$. Consider a given arrival vector $\mathcal{A}$ with $A$ being the total number of online arrivals. By definition, we have $\mathsf{ALG}(\mathcal{A}) = 1$ when $A = 0$.

Now, we show that $\mathsf{ALG}(\mathcal{A}) = b/K$ when (1) $0 < A \leq K$ and (2) $b/K \leq 1$. Note that by dependent rounding, we have (**P1**) $\Pr[Y_j = 1] = b/K$ for all $j \in [K] := \{1, 2, \ldots, K\}$ and (**P2**) $\Pr\left[\sum_{j=1}^{K} Y_j \leq \sum_{j=1}^{K} b/K = b\right] = 1$. Focus on a given $j \in J$ with $A_j > 0$. Consider a specific online arrival of type $j$, which is counted as the $k$th arrival among all online arrivals. When $A \leq K$, we see that $k \leq K$ and the single offline agent will not reach the capacity upon arrival due to (**P2**). Thus, we claim that the type-$j$ agent will be served with probability equal to $\Pr[Y_k = 1] = b/K$ for each of its $A_j$ arrivals. Thus, $\mathbb{E}[X_j] = A_j \cdot b/K$ and $\mathsf{ALG}(\mathcal{A}) = \min_{j:A_j>0}\mathbb{E}_{\mathsf{ALG}}[X_j]/A_j = b/K$. Consider the following three cases.

(**Case 1**) $b > \lambda$. In this case, $K = b$. If $A = 0$, $\mathsf{ALG}(\mathcal{A}) = 1$ and if $0 < A \leq K$, $\mathsf{ALG}(\mathcal{A}) = b/K = 1$. Thus, we claim that $\mathsf{ALG}(\mathcal{A}) = 1$ when $A \leq K$.

$$\mathsf{ALG}(\mathcal{I}) = \mathbb{E}_{\mathcal{A}}[\mathsf{ALG}(\mathcal{A})] \geq \Pr[A \leq K] = 1 - \Pr[\mathrm{Pois}(\lambda) > b] \geq 1 - \exp\left(-\lambda(b/\lambda - 1)^2/(2b/\lambda)\right).$$

Note that $\mathsf{OPT}(\mathcal{I}) \leq 1$. Thus, $\mathsf{ALG}(\mathcal{I})/\mathsf{OPT}(\mathcal{I}) \geq \mathsf{ALG}(\mathcal{I})$ and we are done.

(**Case 2**) $b = \lambda$. In this case, when $A \leq K$, $\mathsf{ALG}(\mathcal{A}) \geq b/K \geq 1/(1+\epsilon)$. Thus,

$$\mathsf{ALG}(\mathcal{I})/\mathsf{OPT}(\mathcal{I}) \geq \mathsf{ALG}(\mathcal{I}) = \mathbb{E}_{\mathcal{A}}[\mathsf{ALG}(\mathcal{A})] \geq \frac{\Pr[A \leq K]}{1+\epsilon} \geq \left(1 - \exp\left(-\frac{\lambda\epsilon^2}{2(1+\epsilon)}\right)\right) \cdot \frac{1}{1+\epsilon}.$$

Since $\epsilon = \sqrt{\ln\lambda/\lambda}$, we establish our claim.

(**Case 3**) $b < \lambda$. We have $\mathsf{ALG}(\mathcal{A}) \geq b/K \geq (b/\lambda)/(1+\epsilon)$ when $A \leq K$. Thus,

$$\mathsf{ALG}(\mathcal{I}) = \mathbb{E}_{\mathcal{A}}[\mathsf{ALG}(\mathcal{A})] \geq \Pr[A \leq K] \cdot \frac{b/\lambda}{1+\epsilon} \geq \left(1 - \exp\left(-\frac{\lambda\epsilon^2}{2(1+\epsilon)}\right)\right) \cdot \frac{b/\lambda}{1+\epsilon}.$$

By Lemma 8, we have that for any given instance $\mathcal{I}(b, \lambda)$ with $b < \lambda$,

$$\frac{\mathsf{ALG}(\mathcal{I})}{\mathsf{OPT}(\mathcal{I})} \geq \left(1 - \exp\left(-\frac{\lambda\epsilon^2}{2(1+\epsilon)}\right)\right) \cdot \frac{1}{1+\epsilon} \cdot \frac{1}{1+1/\lambda+o(1/\lambda)}.$$

Since $\epsilon = \sqrt{\ln \lambda/\lambda}$, we establish our claim. $\qquad\square$

