# OpenReview forum: "Promoting Fairness Among Dynamic Agents in Online-Matching Markets under Known Stationary Arrival Distributions"
_NeurIPS.cc/2024/Conference — NeurIPS 2024 poster_

### Official Review · Reviewer_GfYh · 2024-06-23

**Soundness:** 3
**Presentation:** 3
**Contribution:** 3
**Rating:** 7
**Confidence:** 3

**Summary:**

This paper studies an online bipartite matching problem where each offline type has a capacity and the nodes of each online type arrive according to a Poisson process. Each offline type can serve a certain subset of online types, and each online node needs to be served or discarded immediately upon its arrival. The objective is to maximize individual fairness across all online types, i.e., the minimum matching rate across all online types.

When there is only one offline type, the paper shows that the first-come-first-serve algorithm is $1$-competitive. When there are multiple offline types, the paper proposes an algorithm that is $(1-1/e)$-competitive, whose competitive ratio asymptotically converges to $1$ in several regimes: (1) all offline types have sufficiently large capacities, (2) all online types have sufficiently large arrival rates, or (3) the total offline type capacity and the total online type arrivals are sufficiently unbalanced.

The paper complements its positive results by proving the following negative results. Firstly, no randomized algorithm that matches every arriving online node whenever possible can achieve a competitive ratio better than $1/2$. Next, no randomized algorithm can achieve a competitive ratio better than $\sqrt{3} - 1$. Finally, the $(1-1/e)$-competitiveness of their proposed algorithm is tight.

In addition, this paper generalizes the individual fairness metric to a group-level fairness metric, where agents are divided into multiple groups, groups are allowed to overlap, and the objective becomes maximizing the matching rates across all groups. For this fairness metric, this paper gives two algorithms whose competitive ratio converges to $1$ when offline types' capacities and online types' arrival rates tend to infinity, respectively. This paper also proposes short-run fairness as another individual fairness metric, which resembles ex-post fairness. For short-run fairness, this paper also presents various positive and negative results.

**Strengths:**

This paper studies a well-motivated and interesting problem. Also, this paper is well-written and well-structured in general.

This paper is conceptually strong. In particular, it initializes the study of fairness with respect to online agents and characterizes some special features underlying this model, whereas all prior papers focus on fairness with respect to offline agents. In addition, this paper proposes various fairness metrics that are interesting on their own.

The techniques in this paper are non-trivial and seem correct.

**Weaknesses:**

Some results in the paper are not very promising. For example, the $1$-competitiveness of the FCFS algorithm when there is one offline type seems immediate, and the bounds for short-run fairness and group-level fairness seem quite preliminary.

Minor:
- Line 92: $s^*$ is not previously defined.

**Questions:**

(1) You claim that all hardness results provided in the paper are independent of any benchmarks. However, this is neither explicit in the proof nor discussed in detail. Can you further elaborate on this point?

(2) In Line 308, you claim that there must exist a rare type $t$ for which $E[P_t] \leq (n + 1) / 2$. This is not obvious to me. Can you further explain why this is true?

(3) Do you believe that the $(1-1/e)$-competitiveness given by the SAMP algorithm is tight for this problem?

**Limitations:**

There is no notable limitation.

---

> ### Author Rebuttal · Authors · 2024-08-05
>
> Q1.  When we state that all hardness results provided in the paper are independent of any benchmarks, we mean that all competitiveness results are computed directly against the performance of a clairvoyant optimal policy (OPT), rather than any upper bound on OPT (e.g., the optimal value of a benchmark LP, as claimed in Lemma 1). In many cases, computing the exact performance of OPT is challenging, so we instead compute the optimal value of a benchmark LP, denoted by $\mathsf{Val}(LP)$, as a proxy for OPT.   For any negative results, the upper bound on the competitiveness obtained using $\mathsf{Val}(LP)$ might not accurately reflect the true upper bound based on OPT, as the former is only a lower bound for the latter. In such cases, we must explicitly state the specific benchmark LP used.
>
> Consider the first claim in Theorem 2, for example, which states "Any non-rejecting algorithm (possibly randomized) that serves an incoming agent whenever possible is no more than $1/2$-competitive.'' To prove this claim, a typical approach is to construct an instance and show that the ratio of the performance of the best non-rejecting algorithm to that of a clairvoyant optimal policy is no more than $1/2$. This is exactly what we did: We first created a bad example (Example 1 on page 8) and then showed that (1) the clairvoyant optimal policy achieves a (long-run) fairness of $1-O(1/n)$ (see lines 292 to 296) and (2) any non-rejecting policy achieves a (long-run) fairness of no more than $1/2+O(1/n)$, as shown in the proof of Lemma 5. **Note that we do not compute the optimal value of the benchmark LP (2) to (5) and use it to compute the competitiveness achieved by any non-rejecting policy.** This is why we claim that the upper bound of $1/2$ for non-rejecting algorithms is independent of any benchmark. The same applies to the second and third claims in Theorem 2.
>
>
> Q2. Recall that for each rare type $t=1,2,\ldots,n$, $P_t \in \{1,2,\ldots,n\}$ denotes the position of type $t$ in a randomized order adopted by a non-rejecting policy. Thus, we claim that $\sum_{t=1}^n \mathbb{E}[P_t]=n(n+1)/2$. This implies that at least one rare type $t$ must satisfy $\mathbb{E}[P_t] \leq (n+1)/2$.
>
> We acknowledge that the proof of Lemma 5 lacks sufficient technical details and is therefore not straightforward to follow. In response, we provide an alternative proof for a stronger version of Lemma 5, which states as follows:
>
> **For Example 1, the optimal non-rejecting policy ($\mathsf{Non\text{-}Rej}$) achieves a long-run fairness of $1/2+o(1)$ and $1-o(1)$ for the rare and common types, respectively, where $o(1)$ is a vanishing term when $n \to \infty$.**
>
> The alternative proof, though more technically involved, provides a foundational framework for analyzing a much broader class of online policies and is more self-explanatory. For your interest, please refer to the first section in this PDF (both link and PDF are anonymized): https://drive.google.com/file/d/1Hi_pvwyIbf-aWQuz4W_p7OJU1dnj2JiI/view?usp=sharing.
>
> Q3. Yes. We state it explicitly in the third claim of Theorem 2; see the line 135 and the related proof in Section 5.3 on page 9.

---

> > ### Comment · Reviewer_GfYh · 2024-08-10
> >
> > Thanks for your reply. Please add the provided technical details to the next version of the paper. I have decided to maintain my score.

---

### Official Review · Reviewer_jQss · 2024-07-11

**Soundness:** 4
**Presentation:** 3
**Contribution:** 3
**Rating:** 5
**Confidence:** 3

**Summary:**

This paper considers the online matching problem under known stationary arrival distributions. Each offline agent of type i has a matching capacity b_i, and each online agent of type j arrives according to an independent Poisson process of rate \lambda_j. The objective is to maximize the minimum matching rate among all types of online agents. To defne fairness, the authors introduce three notions FAIR-L (long-run), FAIR-L(G) (group-level), and FAIR-S (short-run).

The authors show that for FAIR-L with multiple offline agents, the SAMP algorithm reaches a 1-1/e competitive ratio, which is a tight ratio for SAMP. Further, the authors provide a counterexample to show that any non-rejection algorithm cannot exceed 1/2 competitive ratio and any algorithm cannot exceed \sqrt{3}-1. The authors also establish bounds for group-level and short-run fairness.

**Strengths:**

1. Conceptually, I think the research direction of incorporating fairness in the online matching problem to be interesting. The fairness notions defined in the paper are natural to me.

2. The results are clean and technically-involved. The competitive ratio for SAMP is tight.

**Weaknesses:**

1. Although the proofs of the theoretical results are non-trivial, they mostly use existing techniques that are common in online matching. The algorithms and their analysis are standard. The overall technical contribution of this paper, in my opinion, is incremental.

**Questions:**

No question.

**Limitations:**

N.A.

---

> ### Author Rebuttal · Authors · 2024-08-06
>
> Thanks for your comments. As for the technical aspects, we want to highlight our parameter-dependent competitive analysis for sampling-based policies. Specifically, we incorporate two parameters—the optimal LP value and the minimum offline serving capacity—into the analysis and the final competitive ratio. In contrast, most existing studies opt for a parameter-free analysis, neglecting the potential impact of different input parameters on the final competitiveness. Additionally, we explore various conditions under which our algorithm can exceed $1-1/e$ or even approach 1 and discuss their practical implications in detail. We believe our parameter-dependent competitive analysis provides a more comprehensive picture of our algorithm's performance across different real-world settings compared to traditional parameter-free analysis.

---

> > ### Comment · Reviewer_jQss · 2024-08-11
> >
> > Thanks a lot for the clarification. I overall like this paper, and I have no further questions.

---

### Official Review · Reviewer_mDTW · 2024-07-12

**Soundness:** 4
**Presentation:** 4
**Contribution:** 3
**Rating:** 7
**Confidence:** 3

**Summary:**

This paper investigates a variant of the online matching problem, focusing on maintaining long-term fairness at both individual and group levels. A key finding is that the optimal competitive ratio achievable without rejecting any items is capped at 1/2. By implementing a novel sampling algorithm, SAMP, the authors achieve a competitive ratio of 1-1/e, with a matching upper bound demonstrated (for the algorithm specifically), and establish that no randomized algorithm can exceed a competitive ratio of sqrt(3) - 1 under their model.

**Strengths:**

- The paper is well-written and provides clear and intuitive definitions of fairness, extending these concepts to accommodate both individual and group dynamics among online items.
- The logical progression from LP-based algorithmic strategies to solid proof constructions aids comprehension and follows a coherent flow.
- The algorithmic results are robust, supplemented by nearly matching upper bounds, suggesting a comprehensive study of the new matching variant given the model and assumptions used.

**Weaknesses:**

- The proof techniques and impossibility results closely resemble those in prior works, potentially limiting the contribution of novel technical methods to the field. However, the intuitive nature of the solutions remains a strength, though it might not significantly expand the community's technical tools.
- While the focus on theoretical analysis is strong, the inclusion of computational experiments could enhance the understanding of these algorithms' practical implications and performance in real-world settings.

**Questions:**

- For the First Come, First Serve (FCFS) for Online Matching with Long-run Fairness (OM-LF) we get a maximal matching thus it achieves a 1/2 approximation ratio in general? (ie the known result of the original online matching problem)
- An exploration into the number of items that must be rejected to surpass the 1/2 competitive ratio would be insightful. Establishing a quantifiable trade-off between the approximation ratio and the number of rejected items could open new avenues for future research.

**Limitations:**

The study's primary focus on theoretical outcomes, while rigorous, leaves some questions about the practical applicability of the algorithms unaddressed. Experimental validation, even if preliminary, would provide valuable insights into how these theoretical constructs perform under more variable and realistic conditions.

---

> ### Author Rebuttal · Authors · 2024-08-05
>
> Q1 ``For the First Come, First Serve (FCFS) for Online Matching with Long-run Fairness (OM-LF) we get a maximal matching
> thus it achieves a 1/2 approximation ratio...''
>
> We politely disagree, and explain why in the following two points.
>
> First, the objective of (individual) long-run fairness is defined as the minimum ratio of the expected number of online agents served to that of arrivals among all online types; see definition (1) on page 2. This differs significantly from the objective of maximizing the expected total number of matches. In fact, these two objectives can conflict in some cases. Consider the bad example (Example 1), for instance. As shown in Lemma 5, any non-rejecting policy can be at most 1/2-competitive. This suggests that improving fairness or competitiveness often necessitates intentionally and strategically rejecting some online arriving agents, which may potentially reduce the number of matches made (i.e., a worse performance under the objective of maximizing the expected total number of matches).
>
> Second, the concept of approximation ratio differs substantially from that of competitiveness, as studied here. Consider a given online maximization problem. Competitiveness (or competitive ratio) is defined as the ratio of the performance of an online policy to that of a clairvoyant optimal; see more details in the paragraph titled "Competitive Ratio" on page 2. In contrast, approximation ratio in the context of online maximization is defined as the ratio of the performance of an online policy to that of an optimal online policy. In other words, competitiveness captures the gap in performance between an online policy and a clairvoyant optimal, where the latter has the advantage of accessing all online arrivals before making decisions. In contrast, approximation ratio reflects the gap in performance between an online policy and an optimal online policy,  both subject to real-time decision-making constraints.
>
> Q2. ``An exploration into the number of items that must be rejected to surpass the 1/2 competitive ratio would be insightful...''
>
> Good point. We follow your suggestion and use the bad example (Example 1 on page 8) to conduct a case study on the trade-off between competitiveness and the number of rejected arriving agents. We hope this case study can provide valuable insights for a comprehensive study of this trade-off in general cases. For your interest, please refer to this PDF (both link and PDF are anonymized): https://drive.google.com/file/d/1Hi_pvwyIbf-aWQuz4W_p7OJU1dnj2JiI/view?usp=sharing.
>
> A few takeaways:
>
> 1) In the first section, we offer an alternative proof for a stronger version of Lemma 5, which states as follows:
>
> **For Example 1, the optimal non-rejecting policy ($\mathsf{Non\text{-}Rej}$) achieves a long-run fairness of $1/2+o(1)$ and $1-o(1)$ for the rare and common types, respectively, where $o(1)$ is a vanishing term when $n \to \infty$.**
>
> The alternative proof, though more technically involved than the existing one, provides a foundational framework for analyzing a general class of online policies.
>
> 2) In the second section, we introduce a randomized rejecting policy parameterized by $\alpha \in [0,1]$, denoted by $\mathsf{Rej}(\alpha)$, which rejects each arriving common-type agent with probability $1-\alpha$, regardless of whether a server is available. We then prove the following lemma:
>
> **The policy $\mathsf{Rej}(\alpha)$ with $\alpha \in [0,1]$ achieves a long-run fairness of $1 - \alpha/2 + o(1)$ and $\alpha - o(1)$ for the rare and common types, respectively, where $o(1)$ is a vanishing term as $n \to \infty$. Additionally, it rejects at least $(1-\alpha)(n-1)$ arriving common-type agents in expectation.**

---

> > ### Comment · Reviewer_mDTW · 2024-08-07
> >
> > No need to be polite, your explanation very well clarifies my question! Thanks for the thorough response and provided supplementary materials. I'm happy to raise my score and hope to see this paper at the upcoming conference.

---

### Official Review · Reviewer_h2oi · 2024-07-14

**Soundness:** 3
**Presentation:** 2
**Contribution:** 4
**Rating:** 6
**Confidence:** 3

**Summary:**

This work focuses on a fair online bipartite matching problem under poisson arrivals, where the objective is to maximize the minimum number of matches across groups of the online nodes (a max min objective). It is showed that if rejecting online nodes is not allowed, then any algorithm will have at best a $1/2$ competitive ratio compared to an omniscient oracle. Even when rejecting is allowed, no randomized algorithm can do better than $\sqrt{3}-1$. Finally, they propose an algorithm and prove corresponding guarantees that depend on the Poisson rates, which achieves in the worst case a $1-1/e$ competitive ratio, and the analysis of the algorithm is tight. Some additional results are given for a variation of the fairness objective.

**Strengths:**

1) The model considered in this paper is a nice addition to the literature of online matching with fairness concerns, the presentation of the model, the algorithms, and their guarantee is clear.
2) The theoretical results are broad, covering lower and upper bound for two different metrics, and solid, with non-trivial analysis. They contribute significantly to the literature on online matching, fairness, and online algorithms.

**Weaknesses:**

While the description of the model and results is clear, the organization and motivation of the paper could be improved.
1) The paper starts directly with the model without discussing why this problem matters, which is especially crucial when dealing with fairness questions. Some mentions of ride-sharing are made on page $6$, but without discussing the connection with the model. It is not explicitly stated what kind of fairness notion is considered in this paper, other than using a max-min objective.
2) There are no references to other works until a remark on page $4$. In particular, some works such as [36] which **specifically** deal with online matching with fairness considerations is only briefly mentioned as pertaining to online matching, not to fair online matching, and no comparisons with it are made. Overall, the position of this work in the existing literature of fairness and fair online matching should be more detailed. See also question $3$.

**Questions:**

Questions:
1) It is mentionned in the related work that Huang et al showed that similar results can be obtained in the KIID and Poisson arrival models. To what extent can the results from this paper be translated to the KIID setting ? Does the same mild assumptions allow to translate results from one model to another ?
2) I am confused about the inherent differences between the long-term "individual" fairness, and the group fairness (6). Which one is harder to achieve? If all the protected groups were non overlapping, is it correct that there would not be any difference in the guarantees provided by the algorithms (by grouping the relevant $\lambda_j$)? When overlapping groups are allowed, why do the guarantees not depend on the group structure, does it consider in some sense the worst-case group overlap? It might be relevant (this is only a suggestion) to discuss briefly connections with the notion of intersectional fairness.
3) There should be a more involved discussion regarding the paper [28] which seems to consider the converse problem of fairness among offline agents. In what ways are the results different? And why is it a more difficult / easier /as hard problem to consider fairness among online agents ?
4) This is simply a curiosity in case the authors are familiar with the prophet secretary literature (if not this question can be ignored), in "Prophet Secretary Through Blind Strategies" Correa et al. proved a $\sqrt{3}-1$ upper bound on the competitive ratio of their problem. Are there any connection between the two or is the fact that the same upper bound was obtained in this paper purely a coincidence?

line 143: $\mathcal{G}=\\{g\\}$ this notation feels a bit self referential.

Small typos : line 98 : outputing -> outputting line 240 : Onine -> Online

**Limitations:**

See weakness

---

> ### Author Rebuttal · Authors · 2024-08-05
>
> Q1 ``It is mentionned in the related work that Huang et al showed that similar results can be obtained in the KIID and
> Poisson arrival models...''
>
> Yes. As stated in the paper by Huang et al., for any algorithm, its competitiveness can be translated between the two models up to a multiplicative factor of $1-O(\Lambda^{-1/2})$, where $\Lambda := \sum_{j \in J} \lambda_j$ represents the total arrival rate of all online types. It is a common practice to consider $\Lambda \to \infty$ or $n \to \infty$ (the counterpart of $\Lambda$ in the KIID setting) in competitive analysis for online matching models under known distributions; see citations such as [14, 18, 31, 22], where all studies adopt this assumption.
>
> Q2. ``I am confused about the inherent differences between the long-term individual'' fairness, and the group fairness...''
>
> Good point! Since individual fairness can be viewed as a special case of group-level fairness, the latter is indeed at least as challenging. The competitiveness result stated in Theorem 3 is evaluated under the worst-possible group structure, as you conjectured. In other words, the competitiveness result could potentially be significantly improved by imposing assumptions on the group structure (e.g., the maximum number of online types per group). As you might expect, overlaps among groups can potentially doom classical policies, such as first-come-first-serve (FCFS), even under very simple settings. Proposition 1 on page 3 states that FCFS is 1-competitive (i.e., matching the performance of a clairvoyant optimal) for individual long-run fairness when there is a single offline agent. In contrast, we can show that **FCFS is zero-competitive for group-level long-run fairness, as defined in (6), under the same setting of a single offline agent**.
>
> Consider the following example: There is a single server with unit capacity. There are $n+1$ online types, indexed as $j=0,1,2,\ldots,n$, each with an arrival rate of 1, and $n$ groups such that each group $k=1,2,\ldots, n$ consists of two types $(0,j)$ with $j=k$. We can verify the following: (1) Any clairvoyant optimal ($\mathsf{OPT}$) can achieve a group-level (long-run) fairness of at least $(1-1/e)/2$. For any offline policy prioritizing serving arriving online types of $j=0$, it achieves a group-level fairness of at least $(1-1/e)/2$. (2) FCFS achieves a group-level fairness of $1/(n+1)$: Note that each group has one agent served by FCFS only when the first arriving agent belongs to one of the two types in that group, which occurs with probability $2/(n+1)$. Thus, we conclude that FCFS is zero-competitive for group-level fairness (when $n \to \infty$).
>
> Based on the above insights into the fundamental differences between the two notions, we propose a different benchmark LP specifically for the group-level fairness and explain the missing role of the optimal LP value ($s^*$) in the final competitiveness; see details in Section D of the paper.
>
> Q3 ``There should be a more involved discussion regarding the paper [28]...''
>
> We offer a more detailed comparison of the two models below.
>
> **Model:** As pointed out on lines 155 to 162, the model in [28] assumes integral arrival rates for all online types and then further assumes, without loss of generality, that each online type has a unit arrival rate. In contrast, we do not make that assumption. Importantly, the integral-arrival-rate assumption among online types allows them to propose a significantly stronger benchmark LP than ours. Specifically, the extra constraint $\sum_{j \in S} x_{ij} \le 1-e^{-\lambda(S)}= 1-e^{-|S|}$ in the LP [28] is crucial for overcoming the $1-1/e$ barrier for algorithms in [28], where $\lambda(S)$ denotes the total arrival rates among all online types in $S$. As acknowledged by [28], their algorithms cannot surpass $1-1/e$ without that constraint. In our case, while the constraint remains valid, it becomes ineffective compared to its role in [28], which is particularly evident when most online types are rare. For instance, when each online type in $S$ has an arrival rate as small as $1/n^2$, this constraint reduces to $\sum_{j \in S} x_{ij} \le 1-e^{-|S|/n^2}$, where the right-hand side approaches one as $n \to \infty$ regardless of the size of $S$.
>
> **Techniques.** Inspired by the insights above, we can no longer exploit any extra constraint to surpass the $1-1/e$ barrier. Instead, we conduct a parameter-dependent competitive analysis for our sampling-based policies and explore various scenarios where our algorithm can exceed $1-1/e$ or even approach one. Specifically, we incorporate two parameters—the optimal LP value and the minimum offline serving capacity—into the analysis and the final competitive ratio. In contrast, the paper [28] conducted a traditional parameter-free analysis, neglecting the potential impact of different parameters in the input instance on the final competitiveness.
>
> Q4. Thanks for noticing this!  We had indeed been curious about this before, but to the best of our understanding, the two upper bounds come from different constructions and different equations.  Ours is obtained by solving $\max_{\tau} \min(\tau+1/2-\tau^2/2,1-\tau)$.  By contrast, theirs is obtained by solving $\min_a\frac{1+a^2/2}{1+a}$.

---

> > ### Comment · Reviewer_h2oi · 2024-08-09
> >
> > Thank you very much for the clarifications and explanations! The above comment regarding group fairness should be included for clarification. I think that adding some motivation for the problem at the beginning would be nice as to why one should care about fairness and matching.
> >
> > Otherwise I have no further questions !

---

> > > ### Author Response · Authors · 2024-08-10
> > >
> > > Thank you for your valuable suggestions. We agree with your points and will incorporate them into future versions.

---

### Decision · Program_Chairs · 2024-09-25

**Decision:**

Accept (poster)

**Comment:**

The reviewers universally recommended acceptance of this paper and appreciate the model introduced in this paper and also the results and technical contributions. There was also some concern raised by the reviewers that was properly address by the authors. The authors are encourage to apply the concerns raised during rebuttal in the final version of the paper and in particular regarding the application motivation of the model and the comment regarding the group fairness.